# Epigenetic Modifications in the Retinal Pigment Epithelium of the Eye During RPE-Related Regeneration or Retinal Diseases in Vertebrates

**DOI:** 10.3390/biomedicines13071552

**Published:** 2025-06-25

**Authors:** Eleonora Grigoryan, Yuliya Markitantova

**Affiliations:** Koltzov Institute of Developmental Biology, Russian Academy of Sciences, 119334 Moscow, Russia; leonore@mail.ru

**Keywords:** retinal pigment epithelium, epigenetic landscape, opposite strategies of cell conversion, retinal regeneration and epithelial–mesenchymal transition, epigenetic regulation, histones and DNA modifications, methylation, acetylation, extracellular signaling

## Abstract

The retinal pigment epithelium (RPE) is a cellular source of retinal regeneration in lower vertebrates and a cellular source of retinal diseases in mammals, including humans. Both processes are based on a genetic program for the conversion of RPE cells into cells of other phenotypes: neural in the first case and mesenchymal in the second. RPE reprogramming in the neural direction is realized in tailed amphibians and bird embryos in vivo, but in higher vertebrates and humans, this process is realized in vitro. Epigenetic regulation determines the phenotypic plasticity of RPE cells, i.e., their choice of the cell differentiation pathway in animals of different classes. It has been suggested that the implementation of the genetic program for RPE reprogramming into different types of retinal neurons in adult amphibians and birds at the early stages of embryogenesis is conditioned by the specificity of the epigenetic landscape. The retinal RPE-dependent pathologies in mammals are characterized by different epigenetic signatures, and have a shared characteristic: specifically, a deficient epigenetic landscape (dysregulations in DNA methylation and histone modifications). Knowledge of the patterns and features of the epigenetic regulation of RPE cell behavior will allow us to obtain RPE cells that are in demand in medicine, from direct reprogramming with the possibility of epigenetically maintaining the cellular identities to the creation of neuro-regenerative technologies for the replacement therapy of RPE-dependent retinal pathologies in humans.

## 1. Introduction

In adult vertebrates, the RPE is a monolayer of specialized polarized epithelial pigmented cells connected by tight junctions and located between the choroid and retinal photoreceptors. RPE cells are characterized by an efficient system of photoreceptor outer segment phagocytosis and high metabolic activity. The RPE maintains retinal homeostasis, protects the neural retina (NR) from oxidative stress, and produces and secretes growth factors. The choroid supplies the RPE and retinal photoreceptors with essential substances and oxygen. From the side of the choroid, the RPE is in close contact with Bruch’s membrane (BM) [1,2]. Interactions of RPE cells with the choroid and Bruch’s membrane contribute to the formation of the blood-retina barrier [2,3]. The main functions of the RPE are phagocytosis of photoreceptor outer segments, their digestion by lysosomes, and retinoid metabolism, which provides light perception [1,4]. The integrity of RPE cell differentiation in the layer is maintained by the coordinated interaction of the endogenous regulatory systems that ensure stable homeostasis processes and tissue functions [5,6,7,8,9,10].

The RPE cells (RPECs) of vertebrates are capable of altering their cellular phenotype and, despite the evolutionary similarity of cellular organization, demonstrate two oppositely directed strategies of conversion (transdifferentiation) in lower and higher vertebrates, after the integrity of this tissue is compromised (Figure 1). Some amphibian species and avian embryos manifest the ability to regenerate the retina through RPE cell conversion up to the formation of a new NR de novo, after the surgical detachment or removal of the original one [11,12,13,14,15,16,17,18]. The conversion of RPE cells in amphibians in vivo and in birds in vitro involves reprogramming the genome: this is based on switching from the program that ensures epithelial, melanogenic differentiation to a program that leads to neural and glial differentiation [19,20,21,22].

In mammals, including humans after retinal detachment and rupture, RPECs undergo conversion along the mesenchymal pathway, which leads to retinal degenerative diseases, such as proliferative retinopathy [23,24,25,26], age-related macular degeneration (AMD) [27,28], and diabetic retinopathy [29], accompanied by epithelial–mesenchymal transformation (EMT) and subretinal fibrosis [30,31,32].

Long-term studies of RPE cell biology and potential to differentiate into various cell phenotypes in vitro conditions have demonstrated that the RPE has cell-type plasticity and capability of conversion not only to mesenchymal cell lineages but also to neural and epithelial cells [33,34,35,36,37,38,39]. Regulation of RPE cell-type conversions along the neural and mesenchymal pathways is mediated by various dynamic regulatory networks, including transcriptional factors (TFs) and the epigenome [40,41,42].

The process of cell reprogramming that underlies the production of induced pluripotent stem cells (iPSCs), nuclear transfer, or cell fusion is accompanied by a change in the epigenetic landscape. The process is aimed at the conformation of chromatin, making it more accessible to TFs that regulate gene expression. This state of chromatin allows a conversion of the cell phenotype (cellular identity), which occurs when the transcriptional program changes [43,44,45]. Various molecular ways of changing the state of chromatin are known. The key ones are DNA methylation and histone modifications in specific genomic regions that regulate genetic expression in the process of changes in cell differentiation during development and regeneration [46,47,48].

The importance of DNA methylation in regulating gene expression in retinal cells became apparent with the emergence of data on genome-wide changes in retinogenesis during development [49]. Temporal and cell type-specific expression of epigenetic modifiers and their selective interaction with a specific set of TFs ensure the sequential differentiation of retinal cells. Retina-specific epigenetic disorders in the DNA, and not only mutations, may contribute to the pathogenesis of a number of retinal diseases. Thus, irregularities in the DNA demethylation process of gene promoters and enhancers during proliferation and differentiation of retinal cell precursors into photoreceptors may significantly contribute to the development of retinitis pigmentosa (RP) [50]. In the retina, histone post-translational modifications, particularly acetylation and methylation, constitute the most studied epigenetic marks. Genome-wide changes in histone marks, as in DNA methylation, have revealed their key role in regulating gene expression during retinal development [40,51,52] and RPE-related diseases [53]. The role of histone modifications in retinal cells has been revealed by pharmacological or genetic inactivation of enzymes that participate in this process [54]. In addition, changes in histone marks are observed during aging and age-related retinal diseases, suggesting their involvement in disease pathogenesis [55,56].

Many studies have shown that histone and DNA methylation levels largely determine the loss of regenerative abilities in mammals with age [57]. Hence, regeneration may require appropriate histone methylation and DNA methylation statuses, as well as histone acetylation. These epigenetic mechanisms are described in more detail in Section 2. In this review, we have summarized the basic mechanisms involved in histone and DNA modifications during eye pigmented cell reprogramming in lower vertebrates, with an emphasis on acetylation and methylation (Section 3, Section 4 and Section 5), before providing specific examples of the role of RPECs in retinal aging and disease (Section 6). This review is an attempt to analyze the few known most striking instances of epigenetic changes and specific epigenetic landscapes arising in two directly opposed types of events conditioning retinal regeneration and, vice versa, retinal diseases in vertebrates, respectively. For this comparison, we have selected, on the one hand, models of retinal regeneration in amphibians and bird embryos due to a conversion of the RPE phenotype to retinal neurons and glia cells. On the other hand, models of well-known retinal diseases in humans and simulated diseases in higher vertebrates have been analyzed. These diseases include age-related macular degeneration (AMD), as well as diabetic and proliferative forms of retinopathies, which are largely dependent on RPE behavior.

This attempt at comparison has been made in order to identify the main changes in the RPE epigenetic landscape, as well as their common features and fundamental differences in vertebrates. When analyzing the research literature, one can note that there are, on the one hand, relatively few facts obtained on regeneration in lower animals and, on the other hand, there is a considerable abundance of data focusing on an analysis of RPE-related human retinal diseases. Nevertheless, some key common and distinctive epigenetic events are highlighted in this review against the background of analyzing existing information.

## 2. The Main Types of Epigenetic Changes

Epigenetic modifications are changes in gene expression and cellular phenotype that occur without changes in the DNA sequence [58]. Epigenetic signals include DNA and histone modifications, histone variants and positioning, higher-order chromatin structure, and non-histone factors associated with chromatin, including RNA. The main epigenetic modifications are DNA methylation, histone methylation and acetylation, and non-coding RNAs, among them are microRNAs (miRNAs) that can inhibit the translation of a number of genes [59,60]. The chemical modifications of DNA and histones have a significant impact on the availability of genomic regulatory sequences and molecular complexes that control transcription and splicing. Currently, there are a large number of reviews devoted to a range of related topics, such as nucleosome remodeling, 3D chromatin organization, and RNA-mediated gene regulation [61,62,63,64].

### 2.1. Histones as the Key Components in Chromatin Organization and Regulation

Chromatin can be defined as a complex of macromolecules of DNA, RNA, and proteins (histones) that provides the physiological state of the genome. DNA in chromatin in eukaryotes is organized into a system of discrete loop domains attached and interacting with elements of the nuclear framework [65]. The points of DNA attachment to the nuclear framework are localized in sites flanking the 5-terminal regions of genes or gene groups and contain binding sites for regulatory transcription factors. Transcriptional activation and repression of genes or gene groups is carried out discretely (modularly), due to controlled conformational changes in the complex within individual DNA domains.

The genomic DNA of eukaryotic cells is packaged with special proteins, histones, to form protein–DNA complexes called chromatin. Histones are small basic proteins that are highly conserved in all eukaryotes. The histone molecule contains a globular C-terminal portion and an amorphous (without a clearly defined secondary structure) N-terminus. The N-termini of histones protrude beyond the nucleus and can interact with other chromatin proteins. Histones contribute to the compaction of DNA in the nucleus, forming macromolecular structures called nucleosomes. The nucleosome is the basic structural unit of chromatin, which typically consists of a 147 bp DNA base pair fragment wrapped about 1.7 times around an octamer of four core histones (H2A, H2B, H3, and H4) [66]. The tails of the histone proteins project from the nucleosome, and many residues in these tails can be post-translationally modified, influencing all DNA-based processes, including chromatin compaction, nucleosome dynamics, and transcription.

Core histones are tightly packed in a globular region, making them accessible to histone-modifying enzymes [67]. Another protein called linker histone H1 interacts with DNA between nucleosomes. Nucleosomes are separated by a short linker DNA fragment and form a “beads-on-a-string” fiber where DNA can be made accessible to other proteins (euchromatin). This fiber can be further compacted with the help of linker histone H1, making DNA largely inaccessible (heterochromatin) [68]. The function of linker proteins is to compact chromatin into higher-order structures: chromosomes. Specific organization of chromatin allows DNA to be tightly packed, to replicate neatly, and to be distributed among daughter cells during cell divisions [66,69]. In addition to histones of basic types, the minor variants that perform specific functions may be present in the genome [70]. Histone proteins regulate the accessibility of cellular factors to DNA. The role of the histone dosage has previously been shown in DNA damage susceptibility and in the efficiency of DNA repair pathways [71].

Nucleosomes control the dynamic accessibility of chromatin and interact with the transcription machinery, and their positioning defines transcription regulation [72]. The regulatory regions of genes are usually either free of nucleosomes or contain so-called “positioned” nucleosomes with a fixed position relative to the gene sequence [73]. This organization is necessary to ensure access of transcription factors and components of the transcription complex to regulatory sites and promoters [74,75]. Neighboring nucleosomes can be located at different distances from each other; their density and regularity of packing depend on the functional state of the chromatin site [72]. In transcriptionally active regions, chromatin is decompacted. In contrast, in heterochromatinized regions, nucleosomes are separated by stretches of DNA (about 40 bp) and arranged regularly [76,77].

The data on primary cell lines from ENCODE obtained from various human tissues have made it possible to carry out a comprehensive analysis of the human genome. The large-scale study has revealed a special role for DNA sequence in a number of processes: in the competitive interaction between nucleosomes and cis-regulatory elements, in stable transcription maintenance, in the positioning of nucleosomes in exons, and in the repulsion of nucleosomes during transcription termination. The authors suggest that in cells, there are parallel competitive mechanisms that are not accompanied by energy-dependent chromatin remodeling and that predetermine the strict positioning of nucleosomes [78].

In non-dividing (resting) cells, chromatin is in two functional states: euchromatin and heterochromatin, transcriptionally active and inactive chromatin or chromosome regions, respectively. Euchromatin is the region of DNA that is accessible and is in an open state of conformation, due to the weakened packaging state of nucleosomes. These regions are more plastic and contain both transcribed and non-transcribed genes. In contrast, heterochromatin is in a tightly packed state, a condensed form that does not allow transcription factors and chromatin-associated proteins [79]. Open and closed chromatin territories are highly organized functional domains with a defined distribution pattern of epigenetic markers [80].

Two categories of heterochromatin are possible: facultative heterochromatin is a dynamic structure that can be decondensed when genes are turned on [81] and become transcriptionally active at specific developmental stages, during nuclear relocalization, or in a heritable context such as monoallelic gene expression [82]. In contrast, constitutive heterochromatin is generally unchanged in its location in the genome across the cell cycle or during cell differentiation [83] and generally adopts the characteristics of sub-cellular localizations [84,85]. Constitutive heterochromatin is more (but not completely) rigid and locks the telomeric, centromeric, and pericentric regions of the chromosomes [83]. Constitutive heterochromatin is mainly found at telomeres, centromeres, and their adjacent silent regions (sub-telomeres and pericentric regions) [68]. These regions are characterized by high condensation, as well as being highly repetitive, constitutively repressed, and enriched in repressive H3K9me2/3 and H4K20me2/3 histone modifications. These regions also display cytosine methylation at CpG dinucleotides and are often bounded by HP1 [86], which interacts with histone H3 methylated at lysine 9 [87], providing constitutive heterochromatin organization [88]. It is known that heterochromatin is the predominant chromatin state of those DNA sequences that control chromosomal stability and prevent mutations and translocations [68,89,90]. The genome in heterochromatin regions is enriched by repetitive sequences and repressed genes associated with morphogenesis and differentiation (imprinting or X-chromosomal inactivation) [91,92,93]. In light of the current point of view, chromatin function is no longer simply reduced to packaging DNA and thereby regulating transcription. Instead, there is a notion of a dynamic chromatin structure state that controls genome activation and function, thereby influencing cell behavior. The dynamic composition of chromatin at different stages of the cell cycle or during the transition from one cell type to another is regulated through multiple epigenetic mechanisms [92,94].

In addition to histones, chromatin includes non-histone structural proteins, such as the proteins of the HMGB family that are known as the “architectural factors” of chromatin, components of nuclear envelope proteins, topoisomerase II, and others. There is also a group of proteins called chromatin remodeling factors, such as SWI2/SNF2 (BRAHMA). ATP-dependent chromatin remodeling factors are able to move nucleosomes on DNA strands: this is needed for transcription initiation or transcriptional repression [95]. Functionally similar histone proteins have been found in most eukaryotic organisms [96,97,98]. Chromatin-modifying and DNA-binding proteins influence the expression of critical cell cycle regulators, the accessibility of origins for DNA replication, DNA repair, and cell fate choice. Chromatin modifiers manage the cell cycle progression locally by regulating the expression of specific genes and globally by controlling chromatin condensation and chromosome segregation [94]. At the same time, the dynamics of epigenetic chromatin modifications during the progression of the cell cycle are controlled in a cell cycle-dependent manner. The cell cycle provides a correct inheritance of epigenetic chromatin modifications to progenitor cells [99].

### 2.2. Histone and DNA Methylation: General Information

Due to their impact on chromatin structure and DNA accessibility, histones are key regulators of all major chromatin-related processes, including DNA transcription, replication, and repair [100]. Histone modification is a regulatory mechanism that is being actively studied in a wide variety of cellular systems. Among the best-characterized regulators required to maintain cellular identities are the Polycomb group (PcG) and Tri-thorax group (TrxG) protein complexes. The PcG and TrxG proteins assemble into appropriate multimeric complexes and exert opposite gene regulatory functions [101,102].

Histones can be methylated, phosphorylated, ADP-ribosylated, ubiquitinylated, and acetylated on various bases [103,104]. Histone 3 lysine 27 di-/tri-methylation (H3K27me2/3) is catalyzed by the enzymatic subunit SET domain-containing protein, Enhancer of zeste homologue 2 (EZH2), from the Polycomb repressive complex 2 (PRC2) [105]. Polycomb (PcG) and Trithorax (TrxG) group proteins give stable epigenetic memory of silent and active gene expression states and then allow poised states in pluripotent cells [106]. Bistability associated with poised chromatin provides its frequent switch between stable active and silent states under a wide range of conditions (Figure 2).

In contrast to Polycomb (PcG), some of the Trithorax (TrxG) proteins maintain transcriptionally active chromatin by catalyzing the trimethylation of histone H3 on lysine K36 (H3K36me3) and lysine K4 (H3K4me3) at transcriptionally active genes and form a complex in which methylase Mll1 catalyzes H3K4me3 histone modifications [107]. H3K27 methylation is considered to be relatively stable and maintains long-term transcriptional repression. However, lysine demethylases such as JMJD3 (Jomonji domain-containing 3, Kdm6b) and UTX (Kdm6a) specifically demethylate H3K27, which results in the activation of genes associated with animal body patterning, inflammation, and, ultimately, resolution of bivalent domains [108]. However, DNA methylation has now been shown to impart context-dependent functions and regulate diverse aspects of mammalian biology [109]. Studies have shown that DNA methylation can cause changes in chromatin structure, DNA conformation, and DNA stability and can also alter the way DNA interacts with proteins, thereby regulating gene expression [110].

The function of DNA methylation is dependent on CpG dinucleotide density and its precise location within a gene [110] and changes with cellular activities [111]. Therefore, the targeting and function of DNA methylation are tightly controlled and involve multiple regulatory mechanisms. DNA methylation can regulate gene expression by maintaining the silent state of chromatin in time- and tissue-specific ways [112]. The production of methylated forms of cytosine (5-methylcytosine, 5mC) is catalyzed by DNA methyl-transferases (DNMTs). DNMT3a and DNMT3b are known to be incorporated into de novo methylation, while DNMT1 is responsible for maintaining DNA methylation patterns during DNA replication [112].

It was believed that epigenetic changes were static in gene expression regulation, but this view is being reformed now because the epigenetic marks, including DNA methylation, are dynamic. DNA methylation is a most important epigenetic modification, exerted by DNMTs at the 5-position of cytosine residues in CpG di-nucleotides, which are often grouped in clusters as CpG islands [113]. DNA methylation of promoter CpG islands can control gene expression, and hence alterations in these processes affect gene function and cell metabolism [114]. Covalent addition of methyl groups to DNA influences transcription and provides genomic stability [115].

Methylated DNA can undergo demethylation, catalyzed by DNA demethylase enzymes. In contrast, active DNA demethylation is a stepwise process mediated by ten-eleven translocation dioxygenases (TETs) that can sequentially oxidize 5mC to 5-hydroxymethylcytosine (5hmC), 5-formylcytosine (5fC), and 5-carboxylcytosine (5caC). Thereafter, 5fC or 5caC can be excised by thymine DNA glycosylase (TDG), and the resulting apyrimidinic sites can be replaced by unmodified cytosine through base excision repair (BER) [116].

### 2.3. The Role of Epigenetic Modification (Methylation, Acetylation) of DNA and Histones in Tissue Regeneration

Epigenetic modification of DNA and histones is an essential regulator of gene expression in tissue and organ regeneration. While previous research suggests that chromatin modifications are associated with the process of regeneration, the mechanisms and key players of these processes are only partially known [117,118]. Enhancer of zeste 2 polycomb repressive complex 2 subunit (Ezh2) is the catalytic subunit of Polycomb repressive complex 2 (PRC2). Ezh2 is a highly conserved histone H3 lysine 27 (H3K27) methyltransferase with a C-terminal SET domain, which exhibits methyltransferase activity. Treatment with the Ezh2 inhibitor 3-deazaneplanocin A (DZNep) prevented the regeneration of amputated *X. laevis* tadpoles [119].

DNA methylation maintains the methylation pattern in embryogenesis and regeneration and ensures proliferation and progression. DNA methylation at enhancers and promoters is closely linked to transcriptional repression, and the establishment of DNA methylation status depends on the activity of DNMTs [120]. Dnmt1 is also involved in the repression of retrotransposons (mobile elements) through DNA methylation in early development. This mechanism provides additional stability for long-term repression and epigenetic propagation throughout development [121]. The ability of the retrotransposons to integrate into various genomic regions and trigger the isolation of promoters and enhancers disrupts their interactions and also provides a mechanism associated with aberrant chromatin organization. During tissue regeneration, retrotransposon silencing is coupled with stem cell activity and is essential for regeneration processes: Adult stem cells coordinately repress these mobile elements and activate lineage genes. Dysregulation of these processes, also related to risk of mutations, may initiate developmental pathologies, including cancer [122].

There are three families of DNMTs with catalytic activity in mammals: DNMT1, DNMT3A, and DNMT3B. DNMT1 is essential for DNA methylation patterns during cell division. DNMT3a and DNMT3b are responsible for the establishment of new methylation patterns for DNA, known as de novo methylation [123,124]. Shh protein regulates limb pattern formation during embryogenesis. Shh expression is driven by the limb-specific enhancer mammals-fishes-conserved-sequence 1 (MFCS1) [125]. It is known that *X. laevis* tadpoles are capable of complete structural and functional restoration of the limb before metamorphosis. In young animals after metamorphosis, the ability to regenerate only simple cartilaginous spike structures without digits after limb amputation is retained. DNA methylation status analyses revealed that MFCS1 was hypomethylated in *X. laevis* tadpoles and highly methylated in froglets. These findings suggest a link between the suppression of regenerative capacity and the methylation status of MFCS1, but this hypothesis requires confirmation using specific DNMT inhibitors and/or knockout/knockdown approaches [126]. The role of DNA methylation has also been demonstrated in a model of pancreatic regeneration of β-cells after their ablation following metronidazole treatment in transgenic zebrafish. As a result, in wild-type fish, almost all β-cells were ablated, but in Dnmt1 mutant zebrafish, the β-cells exhibited an enhanced capacity for regeneration, in contrast to wild-type fish [127,128]. These findings highlight the requirement for a specific DNA methylation pattern for regeneration.

In a model of cardiac regeneration in transgenic zebrafish, it was shown that cardiomyocytes expressing a mutant version of histone 3-H3.3K27M, which inhibits H3K27me3 catalysis, fail to regenerate. At the same time, wound edge cells showed increased expression of structural genes and formation of prominent sarcomeres. It is obvious that H3K27me3-mediated silencing of structural genes is a prerequisite for zebrafish heart regeneration. In addition, a mutant version of histone 3 (H3.3K27M), in which the lysine (K) at position 27 was substituted for methionine (M), had decreased H3K27me3 modifications and modest increases in H3K27ac modifications. The strategy of epigenetic inhibition of similar structural components in the border zone in the heart after a myocardial infarction is considered a promising approach to stimulate regeneration of heart cells in humans [129]. Epigenetic modifications, such as H3K4me3 and H3K27me3, are also involved in the maintenance of pluripotency [130]. Enrichment of H3K9me3 and H3K27me3 is often observed in heterochromatic regions and plays an important role in the silencing of nearby genes [131,132]. H3K9 methylation blocks the induction of pluripotent stem cells (iPSCs) during fibroblast reprogramming. The activity of H3K27me3-related demethylases and methyltransferases is integral to maintaining cell fates and contributing to regeneration [133]. H3K9me3 impedes the totipotent state of cells from mammalian oocytes through somatic cell nuclear transfer (SCNT) [134]. The role of H3K27me3 in genomic imprinting in early mouse embryogenesis is also well known [135]. The requirement for H3K27me3 demethylation for caudal fin regeneration in zebrafish was identified using specific antisense morpholino oligonucleotides Kdm6b.1 injected into zebrafish embryos at the one-cell stage. Activation of demethylases from the KDM6 family plays an important role in the regulation of this process. Demethylation of H3K27me3 by lysine demethylase 6B/Jmjd3 (Kdm6b), which specifically removes methyl groups from H3K27me3, and by lysine-specific demethylase 6A/Utx (Kdm6a), allows re-expression of its target genes [136]. The role of demethylases from this family has been demonstrated in the lateral line model in zebrafish, which is frequently used for studying the mechanisms of peripheral neuronal innervation of sensory organs and their regeneration and degeneration [137]. Studies using this model have shown that treatment with the selective Kdm6b and Kdm6a inhibitor GSK J4 resulted in suppression of cell proliferation in regenerating neuromasts [138].

Histone acetylation (H3K27ac) and RNA polymerase II phosphorylation (RNAP2-Ser5ph) are equally important mechanisms in the regulation of transcriptionally active chromatin and nucleosome dynamics [139]. Histone acetylation level is maintained by a balance of histone acetyltransferase (HAT) and histone deacetylase (HDAC) activities and plays a pivotal role in regeneration [140,141,142,143]. Transcripts of HDAC1 have been identified in the regenerating tail of the *X. laevis* tadpole. Experiments with pharmacological inhibition of HDACs using trichostatin A (TSA) have shown increased levels of histone H4 acylation, accompanied by inhibition of tail regeneration [144]. The adenoviral overexpression of histone acetyltransferase p300 promoted axonal regeneration in a model of retinal ganglion cell degeneration following optic nerve injury in mice [145]. TSA treatments resulted in the activation of multiple regeneration-associated genes during sensory axon regeneration after spinal cord injury in mice [146]. Thus, an appropriate histone acetylation status largely determines the ability of tissues to regenerate.

The pattern of acetylation of histone H3 lysine 27 (H3K27ac) and transcriptional activation of RNA polymerase II (Pol II) correlate with the activation of specific enhancers [147,148]. In contrast, inactivation of enhancers correlates with histone H3 lysine 27 trimethylation (H3K27me3) and H3K9 di- and trimethylation (H3K9me2/3) [149,150]. A recently developed approach combines Fab-based labeling of endogenous protein modifications with single-molecule tracking to quantify the dynamics of chromatin enriched with histone H3 lysine-27 acetylation (H3K27ac) and RNA polymerase II serine-5 phosphorylation (RNAP2-Ser5ph). This approach revealed that chromatin enriched with these two modifications was generally separated. However, high levels of H3K27ac and RNAP2-Ser5ph are not always present together at the same place and time; rather, each of these marks identifies distinct transcriptionally poised or active sites [139].

## 3. Reprogramming of the Retinal Pigment Epithelial Cells and Its Regulation in Amphibians

Homeostasis and reprogramming of retinal pigment epithelial cells (RPECs), as well as the pathological states of the RPECs, are primarily regulated by genetic programs, including TFs and signaling cascades (e.g., leading to EMT), which are adjusted by epigenetic factors [38,51]. Here, we discuss what is known about RPE regenerative responses and their epigenetic regulation in non-mammalian vertebrates, some of which possess remarkable abilities to regenerate the retina and/or RPE after injury, and in mammals (humans), in which the RPE serves as a source of retinal pathologies.

### 3.1. The Genetic and Epigenetic Regulation of RPE Cell-Type Conversion in Tailed Amphibians

In adult newts (Urodela, family Salamandridae), the RPECs become the source of de novo formation of a functioning neural retina (NR) following the optic nerve transection, surgical total removal, or detachment of the NR. Activated RPECs leave their epithelial layer, dedifferentiate, proliferate, and form a population of RPEC-derived neuroblasts, which then exit the proliferative cycle and acquire the phenotypes of neurons and glia [12,13,18]. This process is under the control of TFs, signaling molecules, and epigenetic mechanisms [18]. After surgery, immune response genes and proto-oncogenes *c-fos*, *c-myc*, and *c-jun* are activated in RPECs [151,152,153].

During RPE cell reprogramming, the expression of the neural stem cell marker Musashi-1 occurs. Activation of the RNA-binding protein Musashi-1 recruits Agronaute 2 (AGO2), which is a key effector of RNA-silencing pathways and a modulator of chromatin remodeling [154]. Differential expression of “developmental” homeobox genes, such as *Pax6*, *Prox1*, *Six3*, *Pitx1*, *Pitx2*, etc., along with tissue-specific genes *RPE65*, *CRBP*, and *Otx2*, was observed at the beginning of RPEC de-differentiation and during the process of conversion [19,155,156,157,158,159]. In dedifferentiating RPECs, mRNA levels of *Pax6*, *Prox1*, and *Six3* were lower than those in the population of RPE-derived proliferating neuroblasts. Taken together, the results indicate that in native and de-differentiating RPECs of adult Urodela, certain components of the molecular genetic profile characteristic of retinal development are active, along with genes responsible for RPE cell specialization [160]. The network of signaling cascades controlling NR regeneration in Urodela includes Fgf, Bmp, Wnt, Shh, and Notch signaling [161,162,163,164]. The role of FGF2 has been investigated in detail [162,165,166,167]. FGF2 performs the evolutionarily conserved mitogenic function and stimulates NR regeneration in other animals [168,169,170].

Clearly, the epigenetic regulation of the RPE cell-type conversion process in Urodela requires close investigation. In RPEC-derived neuroblasts of the retinal regenerate, transcriptional activity of the *Ns* gene encoding the nucleolar protein nucleostemin (GNL3) was detected [166]. Nucleostemin (NS) is known as a highly redox-sensitive protein [171], which is involved in several different processes, including embryonic development, cell self-renewal [172,173], reprogramming, telomere maintenance, and genomic stability of stem and progenitor cells [174,175], and also in tissue regeneration and aging [176]. Nucleostemin expression is characteristic of stem and tumor cells and is one of the factors inducing pluripotency. When stem cells differentiate, nucleostemin expression decreases rapidly prior to cell cycle exit both in vitro and in vivo [177]. NS is among other nucleolar proteins, such as FBL and nucleolin (NCL), which are highly enriched in stem cells and play a key role in stem cell renewal [178,179,180].

Several studies have highlighted roles of nucleostemin in maintaining genome integrity and regulating ribosome and chromatin assembly [181]. Under conditions of retinal organotypic culture, the *Ns* gene shows simultaneous expression with the *fgf2*, which suggests their complicity in the regulation of proliferation of reprogramming cells [166]. To date, only the first steps have been taken in studying the epigenetic profile of reprogramming RPECs, where attention is paid to chromatin remodeling using the model of photo-induced retinal detachment in the newt *Pl. waltl* [182,183]. Studies of epigenetic landscape changes in the RPE cells of mice suggest that during RPE cell-type conversion in amphibians, the expression of “pioneer” TFs and demethylation of regulatory elements of photoreceptor genes are possible [40]. It is possible that in sexually mature but paedomorphic newts, which initially possess a number of juvenile properties [183], the downregulation of the differentiation level necessary for conversion occurring with the participation of signal-response enhancers, including epigenetic ones, does not require significant modification. It is possible that the epigenome of newt RPECs is, in some sense, “ready” for activation and dedifferentiation due to the promoters of TF genes responsible for pluripotency are initially hypomethylated. A similar epigenomic property is known to be inherent in resting Müller glia cells (MG) [184,185]. MG, another potential cellular source of retinal regeneration, are capable of reprogramming into retinal neurons [186,187]. For these cells, it has been shown that initial global hypomethylation is followed by de novo methylation. These processes correlate with changes in the gene expression profile [184].

### 3.2. Epigenetic Regulation of RPE Cell-Type Conversion in Tailless Amphibians

The model of retinal regeneration in tadpoles and adult frogs *of Xenopus laevis* currently has a good prospect of use in research. In *X. laevis* frogs, RPECs are involved in de novo retinal formation [17]. Thanks to the numerous research data available on the genomes of these animals, new genetic approaches have emerged for studying NR regeneration and experimental strategies aimed at promoting regeneration in adult animals [188,189]. *X. laevis* is expected to be used to study the mechanisms governing the reactivation of the expression of “developmental genes”, designated as eye field transcriptional factors (EFTFs). The expression of these key regulators underlies the mechanisms of the RPEC reprogramming process in amphibians and birds [57].

Postulating the evolutionary conservatism of “developmental genes”, similar mechanisms of activation in retinal regeneration are hypothesized. Tailless amphibians, as a model for studying epigenetic mechanisms involved in the regulation of RPEC reprogramming, are also promising for another reason: This phenomenon has been previously well studied at morphological and molecular-genetic levels [14,17]. When the retina is removed and its vascular membrane is preserved, RPECs leave their epithelial layer, migrate inward into the ocular cavity, dedifferentiate, proliferate, and form a population of retinal progenitors on the membrane. Cells of the retinal progenitor population multiply and then undergo differentiation. TFs and signaling pathway components involved in this process were investigated. It was discovered that FGF2 is able to accelerate the process of RPEC conversion under in vitro and in vivo conditions after the removal of the native retina [17,190]. FGF2 can activate the MAPK pathway, which in turn induces RPECs to enter the proliferative phase. In the next step, FGF2, by supporting the expression of Pax6 TFs, promotes the differentiation of RPE-derived proliferating progenitors into specialized retinal cell types [191]. The role of TFs in the specification of NR progenitor cells in RPE transdifferentiation during retinal regeneration in *X. laevis* was studied using *pax6* and *rax* genes as examples. Their expression levels were upregulated in RPE-derived neuroblasts. Retinal homeobox Rx gene knock-down prevented regeneration of the retina in *X. laevis* tadpoles [192,193]. The Rax gene was found to be involved in retinal regeneration in pre-metamorphic *X. laevis* [194]. It was later discovered that the triggering event for retinal regeneration involving RPECs in Xenopus is the upregulation of matrix metalloproteinases (MMPs), as well as factors IL-1β and TNF-α [195]. Data on genome function in the process of RPEC conversion in *X. laevis* form the basis for investigating the epigenetic events involved in its regulation. Thus, the model of retinal regeneration in tadpoles and adults of *X. laevis* is quite promising for use in the analysis of the epigenetic regulation discussed in this article. In this context, the study of regeneration-specific signal-response enhancers and epigenetic modifications of DNA and histones is of particular interest.

## 4. Reprogramming of Iris Pigment Epithelial Cells and Their Regulation

The RPECs are an extension of the iris pigment epithelial cells (IPECs) by their embryonic origin, and topologically, both have similar phenotypes. Therefore, the conversion of RPECs into retinal cells in Urodela is a model that is very close to the model of reprogramming iris pigment epithelial cells (IPECs) into lens cells during their regeneration in the same animals. Moreover, the IPECs, like the RPECs, demonstrate the capacity to switch their phenotype to neural/retinal in vitro [13,20]. The process of pigmented epithelial cell reprogramming in RPECs and IPECs has similar regulatory molecular mechanisms. It is clear that direct extrapolation of the data obtained on the IPECs model is not entirely correct. Nevertheless, studies of both models are in favor of possible similarities between the epigenetic changes taking place in both cases. Thus, the analysis of mechanisms of the different pigment epithelial cells’ reprogramming makes it possible to understand the key factors that dictate the choice of a particular cell-type conversion strategy. For this reason, we also present here the data obtained from the IPECs model.

### 4.1. The Epigenetic Mechanisms of the Reprogramming (Transdifferentiation) of Iris Pigment Epithelial Cells in Urodela

Regeneration of the lost lens is possible thanks to the reprogramming of IPECs into lens fibers in a number of species of tailed amphibians [196,197,198], some species of fish [199], and chicken embryos [11]. Wolffian lens regeneration in newts from IPECs is a classic example of cellular phenotype transdifferentiation. The process has been studied extensively, from the morphology of its individual steps to their molecular mechanisms [198,200,201], as well as in newts of different ages [202] and with repeated lens removal [203]. After lens removal, pigmented epithelial cells of the dorsal iris margin lose the initial characteristics of their phenotype, form a population of dedifferentiating, proliferating cells, which form a lens regenerate (the lens vesicle). Then, the lens progenitor cells follow the path of new differentiation into epithelial cells and lens fibers. Initially terminally differentiated, specialized IPECs express melanogenic differentiation genes, such as tyrosinase, *MMP-115*, and *TRP-1*, the expression of which is inhibited in cells during their conversion [201]. IPE-derived dedifferentiated cells of the lens vesicle are similar to tissue stem cells [204]. This assessment is based not only on the changes in the morphology and proliferative activity against the background of melanin synthesis inhibition in the IPECs cells [205] but also on the data on the induction of pluripotency as a result of activation of TFs, such as *Sox2*, *Klf4*, and *c-myc* [151]. It has also been shown that activation of *six-3* and inhibition of BMP signaling are involved in mechanisms of lens regeneration [206].

The model of lens regeneration from iris pigment epithelial cells (IPECs) in newts (Urodela) was used in a series of studies devoted to the epigenetic mechanisms of cell reprogramming (transdifferentiation). Previous works have been devoted to the study of the process of histone modification during IPECs transdifferentiation [207]. As noted above, covalent modifications in histone tails play a key role in the regulation of a series of epigenetic events: among others, changes in cell differentiation in development [208] and regeneration [209,210]. The research group [207] wondered whether there is a relationship between histone modification and differences between the dorsal (regeneration-competent) and ventral (regeneration-incompetent) regions of the iris. The authors used morphology and double-labeled immune-histochemistry (IHC) techniques on sections obtained from the eyes of *Cynops pyrrhogaster* at the early stages of lens regeneration after lentectomy. The intensity of signals labeling histone modification was correlated with the luminescence of iris cells that did not include BrdU, in cells that have not yet entered the proliferative phase. In dedifferentiating IPECs, histone modification has been differentially regulated together with the activation of genes responsible for the process of cell conversion in which IPECs lose their original phenotype. Although tri-methylated histone H3 lysine 4 (*TriMeH3K4*) and acetylated histone *H4* (*AcH4*) were increased, the acetylated histone H3 lysine 9 (*AcH3K9*) was decreased. Regarding the differences in the dorsal and ventral iris, among the modifications associated with gene repression, only *TriMeH3K27* showed a differential distribution pattern. Although in the dorsal iris (the source of regeneration), *TriMeH3K27* was kept at the same levels after lentectomy, its level was increased in the ventral zone of the iris. In contrast, the levels of DiMeH3K9 and TriMeH3K9 were almost constant in both zones of the irises. The results are correlated with lens regeneration inhibition from the ventral iris [210]. These findings are consistent with the data according to which TriMeH3K27 modification, together with the activity of PcG proteins, can mediate gene silencing in development [211]. In embryonic stem cells (ESCs), most of the genes required for development are modified with TriMeH3K27 and are co-modified with TriMeH3K4 [212]. The histone modifications are thought to keep genes inactivated to prepare them for later activation during embryogenesis and regeneration.

The results generally indicate the switching on of global histone modification that is among the mechanisms of the IPECs phenotype conversion in Urodela [207]. At the same time, it is also pointed out that the investigated mechanism of gene function regulation is certainly not the only one in epigenetic regulation.

The same model, IPE-derived lens regenerate cells, demonstrated the expression of linker histone B4, which is characteristic of early oogenesis, and revealed a change in the histone B4/H1 ratio, with preferential accumulation of B4 [207]. Previous studies in other experimental objects, including zebrafish [213,214] and frogs [215], have shown that B4/H1 is specific to oocytes and early embryos before the onset of zygotic gene expression. Using the methods of cloning of newt *B4* cDNA, vivo-morpholino, immunohistochemistry, and western blot analyses, it was possible to show not only the expression of histone B4 per se but also to demonstrate its significance for IPE-derived cell transdifferentiation [207]. Knockdown of B4 led to cell apoptosis and inhibited both proliferation and lens regeneration in general. It was also found that B4 regulates the expression of a key TF, Pax-6, and a marker of lens differentiation, γ-crystallin. Knockdown of B4 upregulated the expression of nucleostemin during IPEC dedifferentiation. The authors have suggested a role for histone B4 in chromatin decondensation, which allows TFs to interact with the promoter region of genes associated with reprogramming, and/or their selective role with respect to the expression of these TFs [207].

Many nuclear proteins are known to undergo dynamic localization changes between the nucleolus and the nucleoplasm, depending on the cellular state [216]. One of the observations concerns the expression and localization of the *Ns* gene and the encoding protein nucleostemin (NS) [217]. Nucleostemin accumulates in the nuclei of IPECs that are undergoing conversion [217]. After six to seven cell cycles and the growth of the dedifferentiating cell population, IPE-derived lens rudiment cells exit the reproductive cycle and begin to express the set of lens-specific α-, β-, and γ-crystallin genes [218]. The striking changes in gene expression in the described model of cellular reprogramming have prompted a close study of the epigenetic mechanisms regulating this process [219]. The subcellular localization of the NS protein in IPECs during their dedifferentiation in newt (*C. pyrrhogaster*), by immunostaining using an NS antibody, in combination with FISH using 18S rDNA as a probe, was studied. In the original IPECs, the rDNA locus could be detected using this probe, but no accumulation of NS was observed in the nucleoli. Two days prior to IPECs reentering the cell cycle, the expression of NS was activated and it rapidly accumulated in the nucleoli of dedifferentiating IPECs. In dedifferentiating IPECs 6 days after lentectomy, strong NS signals were detected in the nucleoli. Interestingly, the IPECs that accumulated NS in their nucleoli at this time were still pigmented, although they initiated dedifferentiation. Later, NS expression decreased drastically during the period of cell exit from the reproductive cycle and at the beginning of differentiation into lens cells. Taken together, the data suggest that high NS expression plays a role in the regulation of dedifferentiation of newt IPECs, in addition to other molecular participants at the early stages of cellular dedifferentiation in newts, regulating the reversal of the cell state from a differentiated state to a stem cell-like state. It is also possible that NS participates in the control of the cell cycle progression of dedifferentiating IPECs [220]. The morphology of the IPECs conversion process indicates that the architecture of the nucleus of these cells is dynamically changing; in addition, the considerable cell sizes in newts [183,221] allow many structures to be visualized. Original IPECs have a small, shrunken nucleus, but as de-differentiation proceeds, the nucleus enlarges significantly and becomes oval in shape, the nucleus–plasma ratio increases significantly, the intranuclear architecture of chromatin becomes apparent, and the nucleolus enlarges significantly.

Newts are also known to possess a very large genome, having multiple repetitive sequences [222]. The localization of these sequences is associated with specific regions of chromatin, in particular centromeres and rDNA regions—regions of transcription of ribosomal RNA genes [223]. The regions of NS and 18S rDNA localization in the nucleoli of dedifferentiating IPECs have demonstrated their co-localization [220]. The studies of the nucleostemin distribution in iris tissue sections in the newt also gave a detailed description of setting up the immuno-FISH method using those repetitive sequence probes against the whole nucleus.

Besides direct regulation of NS expression, C/EBPα phosphorylation promotes complexation of NS with Brm (Brahma chromatin remodeling complex) or HDAC1. This regulatory complex inhibits the expression of c-Myc, Cdc2, and FoxM1B, after tissue injury, or the expression of c-Myc and FOXM1B, respectively [224]. C/EBPα might regulate NS via E2F suppression. The C/EBPα–Brm complex can downregulate the expression of E2F-dependent genes, which are highly co-enriched with NS in cancer cells [225], demonstrating an indirect mechanism by which C/EBPα might regulate NS via E2F suppression. Age-dependent increases in the C/EBPβ–HDAC1 complex formation are also involved in the mechanisms of GSK3β and SIRT1 promoter repression and impaired cell homeostasis [226]. Genome-wide ChIP-on-chip analysis will provide a more complete picture of chromatin dynamics during reprogramming of the eye’s pigmented cells.

### 4.2. RNA-Based Epigenetic Regulation of IPE-Derived Cell Reprogramming

In addition to histone modifications and DNA methylation, the RNA-based mechanism is also one of the main ways of epigenetic control. microRNAs (miRNAs) are participants in the regulation of many biological processes [227,228]. It is well known that miRNAs are capable of controlling gene expression by targeting complementary sequences in many mRNAs. miRNAs are found in many animals and are highly conserved molecules in the evolutionary lineage from Drosophila to humans [229,230,231]. Only some miRNAs’ roles in the molecular machinery of regulation of IPE-derived cell reprogramming have been identified [232,233]. The mRNA profile was analyzed in cells of the dorsal and ventral irises of unoperated animals compared with the same iris regions on the 8th day after lensectomy: the time of active entry of cells into the S-phase and dedifferentiation. Several microRNAs, piRNAs, and other small RNAs were identified. The most evolutionarily conserved miR-124a was used to study its participation in the process of lens regeneration in newts. The most pronounced difference was that miR-124a was expressed at higher levels (nearly fourfold) in the intact dorsal iris when compared with the regeneration-incompetent intact ventral iris, even though the ventral iris expressed miR-124a [206,232]. It is suggested that the induction of regeneration is determined by fine-tuning the expression level, and not only by the presence/absence of the regulatory factor per se. The targets for miR-124a, such as microphthalmia-associated TF Mitf, retinoid X receptor, retinoic acid receptor γ, SOX9, E2F, retinoblastoma-like protein 1, FGFR2, and chordin, may be participants in the regulation of eye development [206,232]. Expanding research of the larger spectrum of miRNAs using microarray analysis revealed the regulation of several miRNAs (miR181, miR184, miR124, miR204, miR125, and members of the *let-7* family, etc.) between the intact dorsal and ventral irises and between the irises at day 8 of regeneration [233]. Differential expression of these epigenetic factors was detected; some miRNAs were more highly expressed in the intact dorsal iris, while others were more highly expressed in the intact ventral irises. Similar differential regulation was observed in the 8-day irises as well [233].

## 5. Reprogramming of RPE Cells and Its Regulation in Chicken Embryos

The switch of cell types is accompanied by reorganization of the epigenetic landscape. This reorganization drives shifts in the transcriptional program and, finally, in cell identity. Currently, researchers use whole-genome bisulfite sequencing (WGBS) for the analysis of DNA methylation and parallel sequencing, which makes it possible to interpret the correlation of epigenetic changes with changes in the transcription profile. These methods have been applied to describing global epigenetic patterns and their changes in chicken retinal development, allowing correlation of global changes in DNA methylation and differential gene expression during retinogenesis [234].

Continuing the research, the same approach has been applied to studying the NR regeneration process from the RPECs in chicken embryos [235]. It has long been known that chickens are able to regenerate the retina from RPE cells at embryonic stages (E4–4.5) [11]. RPE cells dedifferentiate, proliferate, and form a population of progenitors that then differentiate, acquiring the identity of all the major retinal cell types [235,236,237]. During the reprogramming stage, chicken RPECs express TFs known as pluripotency inducers (sox2, c-myc, and klf4), as well as those belonging to the group of eye field TFs (EFTFs). In this case, suppression of the expression of RPE-specific markers occurs. The same study shows that FGF2 exposure is necessary for the initiation of reprogramming and the completion of RPEC conversion [238]. This information allows us to draw an analogy with the regulatory mechanism of cell conversion in the above two models of eye tissue regeneration in Urodela.

Relatively good knowledge of the natural process and its regulatory mechanisms, in particular the role of FGF2, formed the basis for the study of the epigenetic status of reprogramming RPE cells, namely DNA methylation. The authors analyzed the epigenetic modifications in comparison with the differential gene expression during RPEC reprogramming, using high-resolution and three-dimensional (HR-3D) reconstruction confocal microscopy of histone marks and DNA modifications, as well as WGBS [235]. The results showed differential regulation and genome-wide dynamic changes in DNA and histone modifications related to gene expression changes associated with epigenetic marks. Special attention was paid to DNA methylation and demethylation. At the early stage (up to day 3 p/op), the genes associated with reprogramming appeared to be hypomethylated (as in intact RPE). However, the overall picture changed with the beginning of the formation of the RPEC-derived retina progenitor cell population. DNA demethylation is a driver for chick retina regeneration, regulating developmental genes. Expression of genes encoding pluripotency factors has been revealed to correlate with the “open”, accessible state of chromatin. In this cell population, there were decreased levels of H3K27me3, 5mC, and 5hmC, coinciding with elevated levels of H3K27Ac and 5caC, which indicated active demethylation and, in general, genome-wide changes in the active regulatory landscape. Data indicate significant changes in bivalent chromatin, impaired DNA methylation, and active DNA demethylation in the cells that have been reprogrammed and committed to conversion under the influence of FGF2. Comprehensive analysis of the methylome by WGBS confirmed these results. Differentially methylated regions were found in the promoter regions of genes responsible for the regulation of chromatin organization. In addition, it was shown that TET3 (Tet Methyl cytosine Dioxygenase 3) overexpression was sufficient to reprogram the RPE even in the absence of FGF2 [235]. Overall, the results suggest that the demethylation process and TET3 play a key role in the reprogramming of RPE-derived cells and that TET3 can be considered a novel important factor in promoting retinal regeneration in chicken embryos. When considering regeneration in fish and amphibians, we have mentioned reprogramming with dedifferentiation and acquisition of stemness that resident cells in the pathology focus undergo. Together with ECM remodeling, this creates unique conditions for de novo restoration of structures that are derivatives of the central nervous system (CNS), in particular the retinal tissue [239,240].

Transcriptomic studies have not detected age-related changes in genetic expression in aged neural stem cells (NSCs), although aging affects NSCs directly, as demonstrated by biochemical abnormalities, particularly lysosome dysfunction manifested by impaired protein degradation and intracellular accumulation of protein aggregates [241,242]. Different subpopulations of NSCs in the subventricular zone (V-SVZ) demonstrate cardinal differences in protein homeostasis. Thus, activated NSCs are characterized by the presence of active proteasomes, whereas quiescent neural stem cells (qNSCs) accumulate protein aggregates in large lysosomes. Lysosomal activity of NSCs decreases with age, resulting in the accumulation of protein aggregates and loss of activation ability. Stimulation of lysosome function prevents aging-related decrease in activity of lysosomes [243]. The population of activated hippocampal NSCs in mice decreases significantly with age and demonstrates certain molecular hallmarks of aging, among which there is an increase in tyrosine-protein kinase ABL1 (Abl1) expression [244]. The Abl inhibitor imatinib can partially restore the function of senescent NSCs and slow down their aging process. Changes in NSC niches during human aging are much less studied. At the same time, the majority of researchers believe that adult neurogenesis declines dramatically with age, starting from the first year of postnatal life [245,246,247]. It is obvious that the cell aging program has a direct impact on reparative capacity. However, it is equally obvious that the efficiency of implementation of the latter in mammalian brain pathology is low. A large number of studies are devoted to the development of various approaches to improve reparative regeneration with the help of NSCs. The approaches being developed involve additional activation of NSC survival, proliferation, and migration using various biologically active factors, such as LIF, small molecules (metformin, lithium ions), cell therapy, and so on [248,249,250]. In small mammals, for example rodents, used as experimental models, such approaches sometimes contribute to an effective recovery of lost functions, but the probability of their successful translation to human pathology is minimal.

## 6. Reprogramming of RPE Cells and Its Regulation in Mammals

The biology, properties, and behavior of mammalian RPE cells, including humans, are fairly well studied [5,9,251,252,253]. Special attention to RPE cells is primarily due to the fact that the main severe eye diseases—age-related macular dystrophy (AMD) [254,255,256], proliferative vitreoretinopathy (PVR) [257], diabetic retinopathy (DR) [29], and retinitis pigmentosa (RP) [258]—are associated with abnormalities in the RPE layer [38]. These diseases lead to visual impairment and, in extreme cases, vision loss. In these diseases, the phenotype and behavior of RPECs are altered, and a disruption of topological, trophic, regulatory, and functional connections with neural retinal (NR) photoreceptors also occurs.

Here, it is necessary to briefly characterize these main RPE-associated diseases of the human retina (AMD and variants of retinal dystrophies, such as PVR and DR), in the development and regulation of which epigenetic mechanisms are involved, depending in turn on cell homeostasis disorders [259]. In addition to these diseases, the RPE is responsible for the etiology of RP, a disease that, being a heterogeneous genetic disorder [258], is not discussed in this article. The abovementioned diseases are accompanied by the loss of cell–cell interactions, cell death, and disorders in the functional light perception system, RPE, and NR. AMD affects more than 60% of the population aged 65+ [260]. During AMD, there is a loss of photoreceptors in the macular area, where the light rays are focused on the retina [261]. A distinction is made between “dry” and “wet” AMD. In the dry form of AMD, druses consisting of fats, vitronectin, amyloid proteins, and inflammatory proteins are formed in the macular region, in the RPE layer, and beyond. The presence of pathological inclusions, which separate RPECs from Bruch’s membrane and the choroid, leads to RPE layer disorganization, para-inflammatory reactions, and cell death. These events not only disrupt the functioning of RPECs but, as a result, lead to the death of photoreceptor cells [262]. The wet, exudative form of AMD, also developing in the macular region of the retina, is accompanied by disorders in the network of blood vessels of the choroid. The latter become dysfunctional and leaky, which leads to the presence of free blood, disrupting RPE/NR interaction and affecting light perception [263]. Despite the long history and development of adequate therapies, these efforts to preserve vision are still significant [264,265,266,267,268,269]. PVR, often caused by retinal detachment and its tears, is manifested as the withdrawal of RPE cells outside the layer, their epithelial–mesenchymal transition (EMT), and involvement in the epiretinal membrane (ERM) formation [23,25,30,270]. Proliferative diabetic retinopathy [271] and subretinal fibrosis [272,273] are also associated with EMT and proliferation of RPECs. Pathological processes, including ERM formation, lead to vision loss due to impaired interaction and function of RPE and NR. In RPE-associated retinal diseases, prevention of disorders and the restoration of the normal functional connection between RPE and photoreceptors is of particular importance. In this regard, the search for new therapies based on the knowledge of epigenetic modifications in the etiopathogenesis of these diseases may become a breakthrough. However, only the first steps in this direction have been taken so far. Data on key molecular and epigenetic events in RPECs during RPE-associated diseases are summarized in this section. The main epigenetic events in the RPE cells of lower vertebrates (amphibians and birds), in which the RPE is the source of retinal regeneration (Section 3, Section 4 and Section 5) compared to those in higher vertebrates (mammals) (Section 6), are highlighted in Table 1.

### 6.1. RPE-Related Retinal Disease Development and RPEC Aging

In humans, AMD, PVR, and DR are multifactorial diseases that have distinct pathophysiological mechanisms, but ultimately impact the functionality of RPE cells. The dysfunction of RPE cells can exacerbate these diseases. RPE cells are known to undergo oxidative stress, and in pathologies, they exhibit pro-inflammatory responses, mitochondrial and lysosomal dysfunction [274,275], loss of all mitochondrial electron transport complex (ETC) activities [276], and nucleolus and nucleolar stress [277]. Homeostatic imbalance in RPECs is due to the loss of proteostasis as a result of oxidative stress and senescence [278]. Understanding the molecular and epigenetic mechanisms that drive RPE aging is critical to our fundamental scientific understanding of cellular longevity and to the development of novel, effective therapies for age-related retinal diseases. In eye diseases, inflammation, oxidative stress, and lipid metabolism disorders are the major pathogenic factors. The molecular-genetic factors mediating their action may be closely related to abnormal DNA methylation and are being considered as targets for epigenetic regulation for the therapy [279].

Here, we very briefly review some known facts about the involvement of epigenetic mechanisms in retinal aging. This issue has received considerable attention in the literature, including microRNAs, methylation patterns, histone gene expression profiles, and post-translational modifications (PTMs) in the aging RPE and neural retina [280,281]. Interestingly, a recent report showed that the chromatin state of murine adult (2.5–3-month-old) RPE contains several key genes for retinal ganglion cells (RGCs), bipolar, amacrine, and horizontal cells marked with the PCR2 repressive mark H3K27me3 [40].

A characterization of the transcriptomes of retinal and RPE cell types allows us to outline a therapy strategy for a number of genetically inherited human retinal diseases detected in early progenitor fetal cells. Data on the transcriptomes of individual cells (single-cell RNA sequencing) have provided information on specific cellular “transcriptome landscapes”. This information contributes to the spatial and temporal characterization, molecular and genetic characterization, and dynamics of the emergence and onset of specialized cell types of the vertebrate retina and RPE, identifying both similarities and differences [2,252,253]. A recent comprehensive study of the “transcriptome landscapes” was performed on young (2–3 months) and old mice (20–24 months), using mRNA-seq, Western blot analysis, immunohistochemistry, qPCR, cell proliferation assays, and SA-β-Galactosidase assays [56]. The results show a significant reduction in core and linker histone components H1, H2A, H2B, H3, and H4, along with critical regulatory factors in the HLB, including *Hinfp*, *Npat*, and *Casp8ap2* in the aged mouse RPE/choroid. Additionally, using an in vitro aging model of human RPE (hRPE), downregulation of histone expression has been revealed. Based on the results, the authors believe that in the normal aging process in RPECs, there is histone loss and hypoacetylation [56]. The data provide some insight into RPEC aging processes taking place at the epigenetic level.

Studies in the past decades have demonstrated that nucleolar stress plays an integral role in cellular responses and may trigger distinct behaviors in different cell lineages under damaging stress conditions [282]. Inhibition of nucleolar transcription in the initiation of neuronal apoptosis has been shown [283]. rDNA methylation may precisely predict the aging cell status of different organisms. There is a tightly regulated coordination between rDNA methylation and cell aging [284]. The findings indicate that aging cells exhibit both enhanced rRNA transcription and rDNA hypermethylation. Conflict between the reduced general transcriptional capacity of rDNA caused by rDNA methylation and higher rRNA transcription may contribute to the accumulation of rDNA damage and nucleolar defects and trigger the cellular senescence program [285]. Data from the mammalian system in vivo and in vitro confirm a key role of the nucleolus in metabolic regulation. Ablation of Sirt7 has been reported to alter hepatic lipid metabolism and the development of fatty liver diseases, albeit inconsistent results were obtained by different groups. Sirt7 is mainly distributed in the nucleolus in quiescent cells. Upon nucleolar stress, Sirt7 is released into the nucleoplasm and the cytoplasm, where it may alter the acetylation of metabolic regulators [286]. Identification of the molecular pathways responsible for rDNA methylation, epigenetic modifications of metabolic regulators, histone mRNA and protein degradation, as the underlying mechanisms will contribute to understanding the phenomenon of cell aging and the development of age-related and RPE/choroid-related retinal diseases in humans.

### 6.2. Epigenetic Changes Occurring During Proliferative Vitreoretinopathy

Mammals, including humans, lack the ability to regenerate the retina with the help of RPECs, i.e., through their proliferation and reprogramming [20,287,288]. At the same time, the RPE and the NR are exposed to light and oxidative stress, leading to cellular and molecular changes. These changes include mitochondrial DNA damage, suppression of lysosomal and mitochondrial functions, and lipofuscin accumulation in RPECs, and may lead to localized chronic inflammation and drusen accumulation below the retinal pigment epithelium on Bruch’s membrane after retinal detachment [2,289]. Retinal damage leads to inevitable cell death, as well as to a loss of RPECs and neurons, with the possibility of conversion of the latter into cells of mesenchymal phenotypes not excluded.

The process of RPEC reprogramming during PVR is essentially an epithelial–mesenchymal transition (EMT) [26]. In mammalian and human in vivo models, during EMT after retinal detachment and retinal tear in humans, some RPECs undergo phenotypic changes. RPECs lose polarity and junctional complexes, detach from Bruch’s membrane, reorganize the cytoskeleton, change shape, and acquire the properties of mesenchymal cells [290]. They leave the layer and acquire the ability to migrate and proliferate. On this pathway, cell death and conversion of RPECs toward a phenotype resembling the myofibroblast phenotype are possible. The markers of EMT appearance are the appearance of contractile protein α-SMA in cells, changes in the expression pattern of cytokeratins, deposition of ECM proteins, collagen, and fibronectin [291,292]. When moving outside the retina, RPE-derived cells are exposed to inflammatory cytokines and growth factors contained in the vitreous body and produced by the cells of the surrounding tissues. The list of important regulators of EMT includes TGF-β (master regulator of EMT), PDGF, EGF, FGF, VEGF, CTGF, IGF2, IL-1α and β, IL-2, 3, 6, and 8, TNF-α, adhesion factors such as ICAM-1, and others (Figure 3). Among them, TGF-β, TNF-α, PDGF, IL-6, and IL-8 are considered to be particularly important in this process [293,294,295]. Increased levels of TGF-β [296] and TNF-α [297] have been reported in the vitreous body of PVR patients and correlate with the severity of this disease. RPE-derived cells begin to synthesize ECM (extracellular matrix) molecules and remodel the ECM, participating in the formation of the ERM [298,299,300]. ERM formation and contraction of its cells are responsible for the clinical picture of PVR. This unwelcome wound healing of the retina constitutes the cause of approximately 10% of all retinal detachments [301]. All stages of PVR development and their regulation have been widely described [302,303]. Thus, RPEC EMT in situ represents an aberrant tissue response, a repair attempt that results in PVR progression [24].

In the pathway of these processes, RPECs lose their original niche and are exposed to proinflammatory cytokines, which, in turn, enhance PVR-associated processes. Many of the cytokines, including the major inducers TGF-β and TNF-α, are produced by RPECs along with surrounding cells, including macrophages [301]. TGF-b1 and TNF-a are activated synergistically as EMT progresses, as shown using an in vitro model of human RPE stem cells (RPESCs) [41]. RPESC-derived RPE cells produced fibroblastic, contractile membranes resembling those formed by PVR in vivo [302,303]. The molecular and genetic basis of RPEC EMT, as in other examples of RPE cell-type conversions in animals, includes changes in the expression of functionally significant genes under the control of specific TFs, regulatory signaling systems, and epigenetic factors. The EMT process is accompanied by changes in the expression pattern of TFs of the Snail superfamily: Slug, ZEB1/2, TWIST, GSC [291,304], and others, including TFs accompanying EMT, fibrosis, and oncotransformation [305]. Significant TFs and proteins were identified using bioinformatics and biochemical analysis techniques. This list includes Oct-1, hepatocyte nuclear factor 1 (HNF-1), nuclear transcription factor GATA-1, SMAD3, transcription factor E (TFE), interferon regulatory factor-1 (IRF), HNF3alpha, E2F, CDP, SP3, and others. Specific elements responsible for the transcription of these factors have been found in the promoters of these genes. Their regulatory elements are considered the potential targets for the therapy of retinal pathologies associated with RPECs [306,307].

Studies of the molecular mechanisms of RPECs in their EMT often utilize RPE cells derived from human stem cells (hESCs), embryonic RPE, and the ARPE-19 cell line [308,309,310,311,312]. In these experimental models, the proto-oncogene *FOXM1* has been identified, the overexpression of which “reinforces” the original RPE phenotype, in particular the expression of the premelanosomal protein PMEL17 [313]. FOXM1 is also known to regulate RPE cell proliferation through its association with cell cycle regulator genes [314]. Knockdown of *FOXM1* by siRNA downregulates the expression of positive cell cycle regulators (*CDC5L*, *CDK12*, and *FZR1*), and, vice versa, increases the expression of CDKN1A inhibitor [315]. The mechanism of EMT involves modulation of the expression of Wnt5B and BMP7 signaling proteins by FOXM1. The exogenous recombinant Wnt5B has been shown to significantly reduce the expression of epithelial markers when the epithelial phenotype changes toward the mesenchymal one [291].

miRNAs are involved in the regulation of gene expression associated with PVR development. The differential expression of 754 miRNAs was identified in the vitreous of patients with primary retinal detachment [316]. The study of the expression of miR-143-3p, miR-224-5p, miR-361-5p, miR-452-5p, miR-486-3p, and miR-891a-5p demonstrated their increase with the worsening of PVR grading [316]. Another study showed that miR-302d and miR-93 are both capable of inhibiting TGF-β-mediated VEGFA secretion from ARPE-19 cells by directly targeting the receptor TGFbR2 as well as VEGFA, thus preventing TGF-β-induced EMT of ARPE-19 cells in vitro [317].

The significant differences in the phenotypes of RPEC-derived RPE cells, untreated and treated with cytokines, are associated with the switching of putative active enhancers and promoters. This study identified several TFs that the authors consider regulatory [318]. In addition to canonical EMT factors, the identified TFs (KLF6, SOX17, FOSL1) have not previously been considered as such in this context [41]. The aberrant activation of the RANK–NFATc1 pathway was identified as a key driver involved in PVR progression. Data highlight genome-wide shifts in *cis*-regulatory domains of the genes from the transcriptional regulatory network in RPE cells during PVR progression [55]. The increase in the enhancer activity for a number of key genes in EMT is congruent with increased chromatin accessibility in RPE cells during PVR progression. RUNX1 (Runt-related transcription factor 1) and the Vimentin (marker of fibrosis) gene in RPE cells during PVR exhibited heightened chromatin accessibility and elevated levels of H3K27ac, H3K4me1, and H3K4me3 compared with normal RPE cells. Meanwhile, the other histone modifications remained relatively unchanged [55]. To better characterize these processes, epigenetic and transcriptional events in RPE were investigated after treatment of RPE cells with TGF-b1 and TNF-a. RPE cells demonstrated widespread changes in the epigenome accompanied by an unusual duality toward activation of regulatory elements rather than deactivation. These changes in active chromatin signatures occurred largely at distal enhancers, while promoters were less affected. This is consistent with enhancers serving as primary mediators of phenotype changes in RPE cells. H3K27ac enrichment at enhancers demonstrated a high degree of cell specificity and responsiveness to signaling. Histone deacetylase (HDAC)-mediated epigenetic mechanisms play important roles in controlling EMT [41,319]. HDAC1/2 deacetylate histones H3 and H4, suppress E-cadherin expression and promote EMT. Downregulation of HDAC3 expression increases the acetylation of c-Jun and leads to its degradation, which increases the expression of E-cadherin and decreases the expression of Snail, suppressing EMT. HDAC activation in human RPE cells in vitro by TNF-α+TGF-β leads to EMT and fibrosis. It was also found that the vitamin B3 derivative of vitamin A, nicotinamide (NAM), can promote a mesenchymal-to-epithelial transition (MET) in RPE cells, can restore epithelial identity, and can prevent RPE from transforming into ERM-like membranes [41]. The observed massive reorganization of enhancer pattern is consistent with the RPE undergoing a drastic change in regulatory programs in response to changes in the signaling environment. It is emphasized that the most significant event is a gain of active chromatin marks at many putative enhancer elements associated with actively transcribed genes, and NAM was able to downregulate key transcriptional changes in EMT [41]. The authors concluded that NAM can be considered a potential therapeutic strategy to benefit patients with PVR [41]. Tricostatin A (TSA), a class I and II HDAC inhibitor, suppressed the proliferation of RPE cells by G1 phase cell cycle arrest through suppression of cyclin/CDK/p-Rb and activation of p21 and p27, thus inhibiting EMT [319].

Summarizing the data presented in the chapter, we can assume that the key and main epigenetic changes during PVR are those responsible for the upregulation of TFs that trigger a new program of development (reprogramming) of RPECs in a new, but not desirable, direction for them—mesenchymal (Figure 3). Apparently, downregulation of the expression of genes responsible for the original phenotype and maintaining it should occur simultaneously. In addition, epigenetic changes inevitably create a new landscape characterized by changes in metabolism that can lead to impaired cell homeostasis.

### 6.3. Epigenetic Changes During the Development of Diabetic Retinopathy

The research literature relevant to this section is quite extensive. For this reason, only selected facts are presented here, which, however, give an idea of the direction and nature of epigenetic changes and the formation of a new chronic epigenetic landscape associated with this retinal disease. Alterations in RPE and NR cell metabolism in diabetic retinopathy (DR) are a consequence of diabetes progression. The consequence of retinal cell metabolism abnormalities can be macular edema, angiogenesis, microvascular damage, and proliferative diabetic retinopathy [320,321,322]. It is known that the main driver of diabetes is hyperglycemia. In this regard, the term “hyperglycemic memory” is used, which implies the involvement of epigenetic changes maintained for a long time [323]. DNA responds to surrounding stimuli, in this case hyperglycemia, changes its properties, and alters the epigenetic landscape in retinal cells [323]. Hyperglycemia initiates metabolic abnormalities in the retina, inducing genetic changes in cells that can persist even when hyperglycemic conditions disappear [324,325]. Abnormal DNA methylation, base mismatches, and high methylation of cytosine in mitochondrial DNA (mtDNA) due to hyperglycemia can lead to damage [324]. There are examples of molecular-genetic changes obtained in various animal DR models. Thus, using next-generation sequencing, the retinal cells in diabetic mice showed that during the development of DR, there is a significant change in gene expression [326]. A microarray assay performed on cadaveric retinas of diabetic donor patients revealed differential expression of mitochondrial genes. The function of these genes has been associated with angiogenesis, antioxidant defense, and energy production [327]. However, gene expression changes occurring in DR do not have as significant an effect with respect to the development of this disease compared to the expression of AMD-associated genes (Section 6.4). At the same time, in cases of both DR and AMD, a certain similarity of variants of gene expression changes can be traced [328]. From the point of view of epigenetics, when considering the two above-mentioned pathological processes (DR, AMD) in the human retina, the polymorphism of the *SUV39H2* gene encoding H3K9 histone methyltransferase KMT1B is noteworthy [329]. In general, an epigenetic imbalance is observed during DR in both animal and human retinas (Figure 4).

In streptozotocin-induced diabetic rats, for example, this imbalance is represented by global changes in histone acetylation levels, as well as by the altered expression/activity of HDACs and HATs [330,331]. Mass spectroscopy and liquid chromatography identified histone marks that were significantly altered when compared to histone marks in the retina of normal animals and diabetic mice. The authors identified 266 differentially modified histone peptides, including 48 out of 83 methylation marks with significantly different abundance in retinas of diabetic rats as compared to non-diabetic controls [332]. The dysregulation of histone and DNA modifying enzymes is associated with retinopathy characteristics. These processes lead to changes in the epigenetic landscape and can regulate the expression of genes responsible for DR progression. A number of reviews have addressed these issues [51,280,323,333]. It is considered that changes in the *MMP9* and *SOD2* promoters are likely to be associated with alterations in several epigenetic writers and erasers [334], and with their possible role in mitochondrial dysfunction in DR. Increased histone and DNA methylation at promoters of genes *Keap1* and *POLG*, as well as hypermethylation of mitochondrial DNA, have been identified as factors contributing to the expression of genes associated with hyperglycemia in DR [335]. There are few data obtained individually for RPECs. Upregulation of the histone acetyltransferase gene *Ep300* in the diabetic retina and that of two histone arginine methyltransferases, PRMT1 and PRMT4, in diabetic RPECs is known. The expression of endothelial and ECM factors is thought to exacerbate oxidative stress and apoptosis of RPECs via H3R17 dimethylation [336,337,338]. Downregulation of Sirtuin (SIRT1), a conserved protein NAD(+)-dependent deacetylase, has also been reported in DR models. Downregulation of SIRT1 is associated with vascular abnormalities due to impaired cellular metabolism [339,340,341]. A decrease in SIRT6 acetylation and an associated increase in H3K9 and H3K56 acetylation were observed in the retina of diabetic mice [342]. In the retina of diabetic patients, a decrease in SIRT6 acetylation is supposed to drive VEGF (vascular endothelial growth factor) increase and BDNF (brain-derived neurotrophic factor) decrease [343], which are associated with vascular dysfunction and the downregulation of SIRT6 and NMNAT2 (Nicotinamide Mononucleotide Adenylyltransferase 2) [344]. The animal DR models and cell cultures revealed an increase in expression of components of the PRC2 (Polycomb repressive complex), and repression of miR-200b, leading to excess VEGF production in vitro [51,345]. In a rodent model of diabetes, it has been found that hyperglycemic conditions can lead to significant epigenetic changes in Sod2 in H4K20me3 and H3K9. In retinal cells, in the case of DR, changes in histone modifications have been noted along with changes in the enzyme system responsible for histone and DNA modification [323]. The effects of high glucose levels on monomethyl H3K4 (H3K4me1), dimethyl H3K4 (H3K4me2), and lysine-specific demethylase-1 (LSD1) were quantified for the antioxidant enzyme Sod2 by chromatin immunoprecipitation. Histone methylation of retinal Sod2 plays a critical role in the ongoing progression of DR and the metabolic memory phenomenon associated with the disease. The effect of high glucose in retinal endothelial cells is realized through epigenetic modifications at the Keap1 promoter by SetD7, which facilitates its binding with Sp1 and increases its expression [329]. The progress of DR is associated with dysfunctional mitochondria, which accelerates retinal cell apoptosis [333]. It has been experimentally shown that the disbalance in histone modifications affects the maintenance of mitochondrial homeostasis in the diabetic retina [324].

Research on the role of DNA methylation in the development of DR has shown that in diabetic patients, DNMT1 activity is upregulated [346,347]. Differential DNA methylation has been observed in blood samples from patients with the proliferative form of DR as well [348]. Oxidative stress plays a significant role in the development of DR in addition to hyperglycemia, and mitochondrial DNA hypermethylation is an important factor in the chronic imbalance of mitochondrial homeostasis [349,350]. Diabetic disorders increase the activity of Tet (ten-eleven-translocation family dioxygenases) in the retina, which, in turn, plays a significant role in the upregulation of MMP9, a key participant in mitochondrial damage. A high level of Tet2 binding results in promoter hypo-methylation of MMP9, and then in activation of MMP9 transcription [351]. Hypermethylation at the regulatory region of DNA polymerase gamma (POLG1) leads to impaired replication of the mitochondrial DNA and inevitable apoptosis of retinal vascular cells in DR [346]. Hyperglycemic conditions in diabetes promote the methylation of CpG di-nucleotides forming 5-methylcytosine (5mC). The sustained hypermethylation of the CpG sites at the regulatory region of POLG impaired its binding to the mtDNA, disrupting transcriptional activity [352,353]. In the case of DR in retinal cells, changes in histone modifications were observed along with changes in the enzyme system responsible for histone and DNA modification [324]. Thus, the facts given demonstrate the connection between altered mitochondrial DNA methylation and mitochondrial homeostasis. This suggests that therapeutic use of molecules capable of modulating DNA methylation may help to maintain normal cell homeostasis and thereby prevent DR progression [333].

Even the studies cited here only briefly demonstrate the existing link between histone modifications and DR. The development of these ideas and the increase of information on the issue of epigenetic changes and disturbances of the epigenetic landscape in the case of changes in retinal cell homeostasis (RPECs and NR) in DR may bring closer the understanding of this disease. Though studies on this topic are limited, it is known that epigenetic modifications are protective and may be reversible, which can be used for creating therapeutic tools targeting vascular complications in RPE-dependent pathologies [354,355]. In addition, knowledge of epigenetic regulation helps to find reliable “epigenetic” markers of DR and treatment methods for this disease, taking into account endogenous and exogenous factors involved in the degenerative retinal diseases.

### 6.4. Epigenetic Changes During Age-Related Macular Degeneration

Another RPE-dependent, socially significant disease is age-related macular degeneration (AMD) that causes central visual impairment [333,356,357]. AMD represents one of the most frequent age-related abnormalities, dramatically affecting the quality of life of older adults worldwide. The interplay between various genetic and non-genetic factors, widely studied, is the basis of ideas about the etiopathogenesis of AMD. Nevertheless, a deeper understanding of the molecular machinery associated with the risk, onset, progression, and effectiveness of therapies is still missing. There is therefore a clear need for research into the epigenetic mechanisms involved in the onset and progression of this disease. This section deals with the epigenomic signatures mostly investigated in AMD, which could be applied to improve the knowledge of disease mechanisms and to set up novel or modified treatment options.

In research of AMD, DNA methylation has been investigated as a contributor to the changes in gene expression that accompany the disease [281,358]. The largest genome-wide methylation analysis of the RPE in AMD, along with associated gene expression changes, identified novel gene targets for functional and future therapeutic intervention studies. The novel differentially methylated genes SKI and GTF2H4 regulate the disease pathways involved in AMD, including TGF-β signaling (SKI) and transcription-dependent DNA repair mechanisms (GTF2H4) [281]. The studies were based on the accepted view that a significant, if not the main, role in the emergence of this disease is played by permanent oxidative stress occurring in the course of life due to light exposure [359,360]. In this regard, the work of genes responsible for antioxidant defense and its epigenetic regulation, namely DNA methylation, has been studied. For this purpose, the Infinium Human Methylation Illumina platform for DNA bisulfite sequencing to compare the methylation status in the postmortem RPE/choroid of human AMD patients and in age-matched controls was applied. Coupling expression results from the Affymetrix Exon Array to DNA bisulfite sequencing showed a significant methylation change of promoter CpG sites that corresponded to the altered expression of 63 genes. It was found that mRNA levels of Glutathione S-transferase isoform mu1 (GSTM1) and mu5 (GSTM5) were significantly reduced in AMD when compared with age-matched controls in the RPE/choroid (Figure 5). Overall, comparison of DNA methylation together with mRNA levels showed significant differences between the RPE/choroid and the NR in AMD versus normal retinal tissue. The results of this study constitute evidence that GSTM1 and GSTM5 undergo epigenetic repression in AMD RPE/choroid, which may increase susceptibility to oxidative stress in the AMD retinas [358]. The data allow us to conclude that a global dysfunction or loss of function of epigenetic control mechanisms can be observed within the retinal cells of AMD patients.

Histones serve as the primary carriers of epigenetic information in the form of post-translational modifications that are vital for controlling gene expression, maintaining cell identity, and ensuring proper cellular function. Loss of histones in the aging genome can drastically affect the epigenetic landscape of the cell, resulting in altered chromatin structure and changes in gene expression profiles. The effect of age-related changes on histone levels and histone acetylation in RPE and neural retinal cells in mice has been shown. The authors observed a global decrease in histones H1, H2A, H2B, H3, and H4 in the aging RPE/choroid tissue complex but not in the neural retina. Transcriptome analysis revealed a significant decrease in the expression of genes involved in histone regulation in the aging RPE/choroid, including elements of the histone body locus (HLB) complex involved in histone pre-mRNA processing [56]. The loss of histones in the RPE cells with age in humans has been shown by immunostaining of retinal sections. Knockdown of Histone Nuclear Factor P (HINFP), a key component of HLB, in human RPE cells appeared to cause histone loss, senescence, and upregulation of markers of senescence-associated secretory phenotype (SASP). Replicative senescence in human RPE cells also causes progressive histone loss and SASP acquisition. Acetyl histone profiling in the RPE/choroid of aged mice revealed a specific molecular signature with loss of global acetyl histone levels, including H3K14ac, H3K56ac, and H4K16ac tags [56]. Taken together, the results strongly demonstrate histone loss as a unique feature of RPE aging and provide information on potential mechanisms linking histone dynamics and cellular aging.

## 7. Endogenous Prerequisites Determining Cell Conversion, Epigenetic Regulation

Studies of the epigenetic regulation of RPEC behavior in regeneration and pathologies raise a number of important aspects for the regenerative biology of tissues and organs in vertebrates. Fundamental issues are concerned with the problem of regulation of cell differentiation stability and plasticity, the relationship between regeneration and carcinogenesis, and regeneration and the immune system. These problems have a long history of research.

Cellular reprogramming of RPECs in regeneration-competent species involves dedifferentiation, transdifferentiation through the formation of stem-like neuroblasts, and RPEC redifferentiation. The process of RPEC conversion reutilizes part of the developmental programs to restore the damaged retinal tissue. Recapitulation of the specific developmental steps from the stage of neuroblast production ensures the precise and correct patterning for tissue growth [153]. It is obvious that endogenous systems tightly regulate the number of dividing undifferentiated cells (neuroblasts) and their progenitors in tailed amphibians. This regulation allows avoidance of abnormal cell growth and malignant transformation.

Urodeles are thought to have low rates of tumorigenesis [361]. It has been shown that more regenerative eye cells, such as IPCs, are able to avoid transformation into tumor cells in tailed amphibians [362]. The problem of the relationship between differentiation status, regeneration capacity, and tumor suppression, one of the intriguing questions, is discussed in recent reviews [363,364,365].

The natural regeneration signals can reprogram the chromatin state of the RPECs to stem-like progenitor states [235]. It is assumed that natural RPEC reprogramming and regeneration are possible due to the peculiarities of the chromatin structure.

There is a point of view that the flexible chromatin state at many regeneration-related loci is maintained in some urodele animals throughout development and adulthood [366]. Changes in the expression of histone modifiers, for example, Polycomb group (PcG) and Trithorax group (TrxG) proteins, histone modification, and changes in the extent of DNA methylation during RPEC conversion provide their phenotypic switch through regulation of the complex transcriptional programs [235]. The poised chromatin state is due to synergistic epigenetic regulation by PcG and TrxG proteins [364].

Activation of the immune system after tissue injury is a main prerequisite for healing and regeneration [367]. An inverse relationship is supposed between the maturity of the immune system and the ability to regenerate in vertebrates [368]. The specifics of immune response in caudate amphibians and their long-term healing/regeneration without visible inflammation and scar is another prerequisite for successful cell reprogramming and regeneration. The “primitive” immature immune system inherent in amphibians is an argument for the explanations for their high regenerative ability [369]. Their immune response is adaptive in nature and formed in the course of evolution, playing a regulatory role in regeneration [370]. In particular, the features of the adaptive immune response in tailed amphibians may be manifested in the pattern of MHC variations that arise as a result of the action of evolutionary selection on MHC, although its mechanisms are certainly much more complex [371]. It is important to emphasize that the ancient and primitive features of the amphibian immune system determine its ability not only to regulate cellular regeneration processes but also to take direct part in molecular mechanisms [372].

In mammals, the enhanced control of the cell differentiation status is strictly coupled to the establishment of the complex multilevel control of the mitotic cycle, increasing complexity of the regulation of cyclin-dependent kinases and oncogenes, and explains the decline in regenerative capacity [373]. Unlike amphibians, the chronic tissue damage in mammals is accompanied by persistent inflammation, and oxidative stress can cause DNA damage and epigenetic alterations, and alter DNA methylation patterns and histone modifications, resulting in changes in chromatin structure DNA accessibility and gene expression. These dramatic changes may impair regeneration and promote senescence, even cancer, and the latter is characterized by an aberrant epigenetic landscape.

In mammals, chronic inflammation and accumulation of senescent cells, as it is known, elevate the levels of cytokines that continuously activate signaling pathways leading to both DNA methylation and histone modifications, and may drive extensive epigenetic reprogramming. The process enhances uncontrolled proliferation and fibrogenic pathways, shifting the balance towards fibrosis (an aberrant form of regeneration) and cell degeneration [374]. Thus, chronic inflammatory signals can lead to persistent epigenetic changes that contribute to age-related decline in tissue regeneration and increase cancer risk, with amplification of tumorigenic cells carrying mutations [375]. Interestingly, the embryonic tissues can successfully reprogram cancer cells into better-behaved cells. The ability to reprogram melanoma tumor cells by embryonic microenvironment has been demonstrated using embryonic models of human stem cells, the zebrafish, and the chicken. In this case, a reversion of the metastatic phenotype of aggressive melanoma cells occurred, probably due to the convergence of embryonic and tumor-promoting signaling pathways [376]. It would be important to identify the mechanisms through which cancer might behave in embryonic microenvironment conditions, given the fact that tissue regeneration can recapitulate, to some extent, embryonic microenvironment conditions. Resistance to carcinogenesis during RPE reprogramming in the adult neonate can correlate with upregulation of tumor suppressor genes and be determined by a tightly regulated balance between oncogenes and tumor suppressor genes. Regeneration and carcinogenesis are supposed to be two sides of the same mechanism. There is a hypothesis that in regenerative-competitive animals (salamanders and frogs), regenerative processes revert cancerous cells back to a baseline physiological phenotype, in contrast to species (mammals) with low regenerative capacity. The hypothesis is supported by experimental evidence that in mammalian chronic inflammation, caused by non-healing trauma in tissue, attempted repairs often induce an accumulation of epigenetic lesions that can trigger cancer initiation, mainly in epithelial cells [363]. Developmental changes in the epigenome that are associated with tumor development are central to human disease. The cell-type-specific epigenome of differentiated cells is thought to be relatively stable once established during development. In mice and humans with retinoblastomas, the retinal progenitors are capable of switching from neurogenic to terminal patterns of cell division. The epigenetic changes in retinal cells were more prevalent at differentiation genes than in progenitors in retinal neurogenesis. It is suggested that the oncogenic transformation is most likely to perturb genes epigenetically regulated during retinal development. The most dramatic change was de-repression of cell type-specific differentiation programs. Gene ontology and signaling pathway analysis showed the upregulation of the pathways involved in proliferation, and the most common upregulated pathways were those involved in RNA processing and metabolism [377].

In general, the cell responses to injury, inducing retinal pathologies in mammals, in adult amphibians have species-specific features, providing the successful reprogramming of RPECs and retinal regeneration. Inherent features in tailed amphibians are manifested in the following: rapid hemostasis; recruitment of immune cells and factors of endogenous defense systems; diverse activities of the immature immune system; participation of immune cells in the creation of a pro-regenerative environment; maintaining the balance of early proliferation and late apoptosis between scarring and regenerative phenotypes; efficiency of ECM remodeling; rapid rearrangement of the cytoskeleton and cell surface [153]; decrease in the activity of metabolic processes characteristic of cold-blooded animals and high cell viability [373]; and paedomorphosis with attributes of juvenile traits [378].

The underlined features serve as prerequisites for regeneration and are determined by genetic regulatory networks, including signaling pathways and transcription factors, and by epigenetic machinery and signals from the cellular microenvironment. There is reason to believe that the reparative process in some caudate amphibians relies on molecular and cellular mechanisms capable of sensing abnormal (defective) signals and effectively reprogramming or bypassing these signals. It is likely that the innate immune system in regeneration-competent species performs a pivotal function, maintaining a physiological balance to provide surveillance against initiation or to control tumor initiation and progression. It could be suggested that in caudate amphibians, during the process of RPE cell conversion, the embryonic-like microenvironment and the permissive epigenetic landscape are created (Figure 6).

In general, the molecular-genetic and epigenetic machinery supports RPEC reprogramming and ensures genomic stability and functionality, while minimizing the possibility of cell degeneration, EMT, and malignant transformation in tailed amphibians.

Questions concerning the precise mechanisms of epigenetic regulation of RPE conversion, which ensure successful reprogramming without allowing cancer development, require a detailed study.

## 8. Overall Discussion

Despite the fact that a relatively complete picture is not yet available, it is possible to emphasize the outlined regularities. The main known epigenetic mechanisms have been studied in different models of RPEC conversion. Accumulating data gradually add up to a general picture of epigenetic regulation responsible for regeneration or pathology driven by the behavior of pigmented eye tissues. These mechanisms include chromatin remodeling, global histone remodeling, DNA methylation/demethylation, and also alteration of non-coding RNA signatures. In the brief analysis of pigmented cells at the level of epigenetic regulation of gene expression associated with regeneration and/or pathology of the retina, presented in this review, the similarity of the mechanisms themselves, but a fundamental difference in their operation, is noted. In general terms, these differences in regenerative responses of RPECs in vertebrates are summarized as follows (see Table 1).

One of the key things is the very different strategies and nature of epigenetic regulation. In the case of studies in animals whose pigmented iris and retinal epithelial cells are capable of lens and retinal regeneration, the few known epigenetic changes are similar to those occurring in development. Thus, these animals recruit developmental mechanisms of epigenetic regulation along with and, in the course of utilizing, multipotency genes operated during embryogenesis. Thus, during lens regeneration in Urodela, expression of linker histone B4 and synthesis of the nucleolar protein nucleostemin, characteristic of NSCs, were detected at the stage of reprogramming IPE-derived cells into lens fibers [217]. For convertible RPECs, another example of this kind can be the fact that the chicken embryo recapitulates the patterns of H3K27me3 observed during RGCs differentiation in mouse retina development [235].

In diseases of the RPE and retina in humans, a very different picture of the epigenetic signature is at work. In these cases, a deficient or aberrant character of epigenetic regulatory machinery may take place, making the epigenetic landscape dysfunctional. The loss of epigenetic information is also one of the features of the aging RPE. Epigenetic studies using aging animal models (with features resembling the age-associated retinal diseases in humans) demonstrate the loss of heterochromatin, which is a characteristic of cellular aging and genome instability [379,380,381].

There is limited research on the role of epigenetic changes in the regenerative responses of RPE cells that are the sources of eye tissue regeneration. This concerns both age-related changes and those induced by the cellular microenvironment. In one of the few studies in this field, in the already cited work by Jorstad et al. (2020) [382], a regenerative response of Müller glia cells (MGs) was achieved in the retinas of adult mice whose neurogenic potential had been exhausted. It was found that the age dependence was due to the loss of access for incorporation of the *Ascl1* gene, specifically transfected for MG activation, into the genome due to an age-associated epigenetic factor, which limited regeneration. Using a modern, highly sensitive method of integrative epigenomic analysis revealing closed and open chromatin regions, the authors proved that a histone deacetylase inhibitor increases access to key regions of MG genes and thus promotes the production of MG precursors for retinal regeneration. The latter not only displayed a spectrum of marker proteins of inner retinal cells but also formed synapses with pre-existing neurons and responded to light stimulation [382]. For fish retinal MG cells, evidence of DNA methylation/demethylation landscape changes (DNA methylation landscape) during conversion to poorly differentiated neuronal precursors have been found [184]. This study contributes to the understanding of one of the epigenetic mechanisms regulating cellular conversion to produce poorly differentiated regeneration source cells. The authors of the study proceeded from the following logic. The expression of genes encoding pluripotency factors Oct4, Klf4, Sox2, c-Myc, Lin28, and Nanog correlates with the “open”, accessible state of chromatin. In somatic cells, by contrast, there is repression of pluripotency genes on the background of less accessible, condensed heterochromatin [383]. At the same time, the state of chromatin is significantly dependent on DNA methylation [184,384].

DNA demethylation in promoters of pluripotency genes usually correlates with increased expression during reprogramming of somatic cells into pluripotent cells [385]. Reprogramming of MG cells from goldfish retina revealed that the pattern found in the generation of induced pluripotent cells was not reproduced in this model [184]. Chromatin hypomethylation was observed in the initially differentiated MG cells, as in the first days of MG conversion. Thereafter, there was an increase in the level of methylation. As a result, the authors made a general conclusion that, during reprogramming of MG into progenitor cells for regeneration, the genome indeed undergoes dynamic changes in the level of DNA methylation. At the same time, a feature is DNA hypomethylation in native MG cells, which, according to the authors, indicates their initial “readiness” for conversion [184].

The modification of histones–chromatin-related proteins is another epigenetic mechanism of regulation of RPEC reprogramming (transdifferentiation) during regeneration. Histone modification was studied in newt iris cells in the process of conversion to lens cells. The method of quantitative immunohistochemistry revealed significant changes in this epigenetic parameter in cells undergoing conversion in the region of the pupillary margin of the dorsal iris—the area of lens regeneration [219]. On the same model, the expression of genes encoding histone acetyltransferase and histone deacetylase was detected in the gene expression profile among those considered to be regulators of transdifferentiation. The dynamic balance between them, as it is supposed, can also serve as an epigenetic mechanism of regulation of cell conversion in vivo. In the same study, the expression of the nucleostemin gene (*Ns*), characteristic of stem cells, was detected in newt IPECs. The nucleostemin gene, as well as its product, the nuclear ribosomal protein nucleostemin, was also detected in the eye RPECs of the adult newt *P. waltl* [217]. Nucleostemin has been shown to be multifunctional, and its role in DNA protection and its replicative function is necessary for stem cells in particular [178]. In general, the issue of epigenetic regulation of gene expression in the processes that ensure regeneration, such as reprogramming, proliferation, and cell death, requires further extensive investigation. Currently, nucleolar stress is considered in the context of an increasing number of pathologies associated with degenerative processes, immune responses, and impaired embryonic development [277,386]. Nucleolar stress associated with impaired ribosome biogenesis can lead to stress-associated changes affecting processes such as apoptosis, senescence, and cell autophagy [282,387]. At the same time, nuclear proteins can also play a protective role as components of endogenous cell defense systems [388]. Strategies for the selective activation of core survival and repair functions may be promising for preventing cell death and the development of degenerative processes in tissues. However, many aspects of the participation of nuclear proteins in the mechanisms of cellular conversion, as well as in the context of the contribution of the nucleus in neurodegenerative pathologies, remain incompletely elucidated and require better understanding. Thus, as we can see, the key event of RPE transdifferentiation in amphibians is natural reprogramming. It represents a switch from the genetic program providing epithelial and melanogenic differentiation to the program leading to neural and glial differentiation and RPEC layer restoration [389]. How reprogramming is regulated by the epigenome is a matter of detailed study in the future. Currently, there is an understanding that a significant role in the regulation of the expression of certain genes in this model is played by the dynamic composition of chromatin [390,391,392,393].

The nature of chromatin rearrangements during RPE reprogramming in the adult newt and the role of these rearrangements in activating the transcriptional program that enables retinogenic differentiation have only been tentatively studied. An attempt has been made to characterize the chromatin state at the beginning of newt RPE conversion during RPE and NR detachment induced by bright light irradiation of the retina or by mechanical detachment [182]. The data indicated that in RPE cells displaced from the layer inward on days 7–10 after NR detachment, on stained semi-thin sections (1 μm), there was an evident compactization of chromatin in the nucleus when compared with cells retained in the RPE layer. In the nuclei of cells in the RPE layer, a diffuse distribution of small loci of chromatin compactification near the nuclear membrane (so-called peripheral heterochromatin) was observed. In displaced RPE cells, there was an increase in the volume of compacted chromatin (heterochromatin near the centromeric area). The ratio of condensed and diffuse chromatin changed in favor of the former, its displacement to the center, and folding of the nuclear membrane. All changes were quantitatively assessed using computer software ImageJ (https://imagej.net/ij/download.html (accessed on 5 May 2025)) [182]. The described changes in chromatin structure are not known to involve active transcription but, on the contrary, indicate repression of this process in retinal development and diseases [394]. We hypothesize that the chromatin and nuclear membrane changes we have detected at the beginning of the RPEC reprogramming process in newts [182] are related to the response to cellular stress, allow escape from cell death, and prepare for proliferation [153]. Another characteristic is the peculiarity of the newt genome, having multiple repetitive sequences [222], including those associated with specific regions of chromatin, in particular centromeres and rDNA regions—regions of transcription of ribosomal RNA genes [222,223]. It is known that these regions are essential for chromosome stability and the silencing of neighboring genes. The repetitive sequences are involved in the formation of a constitutive heterochromatin structure that provides protection for cells. Despite the conservatism of the process of heterochromatin formation, its epigenetic regulation is dynamic for species and can vary according to the cell state [91].

In adult newts, in addition to the events associated with chromatin reorganization, it is also known that in the RPE, the expression of nucleostemin encoded by the *Ns* gene takes place [166]. *Ns* is a regulator of transcriptional activity, a nuclear protein characteristic of stem and tumor cells, mentioned above for the model of IPEC-derived lens regeneration [219,220]. It has been suggested that the operation of the *Ns* gene in newt RPE tissue is an epigenetic prerequisite for the high plasticity of differentiation of these cells [166,219]. The study of the epigenetic landscape of native RPE cells of adult mice [40] relied on facts about the properties of retinal progenitors to the RPE and NR, known to develop from the shared neuroepithelium of the optic vesicle [4,395,396].

It has been hypothesized that adult mouse RPE cells may retain some similarity to the epigenetic landscape inherent to retinal precursors in development. The authors [40] found that the state of methylomes in RPE cells of embryos and adult mice is indeed similar: most gene promoters are located in open (active) chromatin, characteristic of epigenetically mobile stem and progenitor cells. Detailed analysis of methylation of genes responsible for the specification of retinal cell phenotypes has indicated the localization of promoters of genes controlling the differentiation of non-photoreceptor neurons in repressed chromatin located in unmethylated (weakly methylated) regions of chromatin. According to the authors [40], activation of these genes is possible in the presence of “pioneer” TFs capable of initiating transcriptional events in closed chromatin. At the same time, most of the genes responsible for photoreceptor differentiation were found to be highly methylated. Thus, demethylation of regulatory elements of photoreceptor genes is another mechanism required for the realization of RPEC conversion in the retinal direction. It is assumed that both mechanisms (activation of pioneer TFs and demethylation of regulatory elements of photoreceptor genes) are intrinsic to the RPE of tailed amphibians and operate during NR regeneration after damage [40].

It is also possible that in newts, which are initially juvenile due to their paedomorphic state [163,183], the lowering of the level of differentiation during reprogramming does not require large-scale epigenetic modifications of the genome and, in general, gene function. One can assume the presence of an epigenetic landscape permissive for rapid switching of the phenotype conversion program of the RPE cells. It has been supposed that in regeneration-competent lower vertebrates (tailed amphibians), the chromatin structure that favors the OFF transcriptional state is characterized by two states: flexible chromatin or more rigidly inactive chromatin. The regeneration-competent lower vertebrates may maintain a flexible chromatin state at many regeneration-related loci throughout ontogeny. Anuran (Xenopus) in vivo probably lose this chromatin state during metamorphosis, and mammals lose this chromatin state early in embryogenesis [209]. According to this hypothesis, the chromatin structure that favors the OFF transcriptional state in RPECs may be characterized by two states: flexible chromatin or more rigidly inactive chromatin (Figure 7).

This assumption can be confirmed by the joint work of genes encoding features of both the original and new differentiation [160]. Indirect evidence of the “flexibility” of the epigenetic landscape even in advanced mammalian RPECs, allowing spontaneous development in the NR, is provided by experiments with sphere-induced RPE stem cells (iRPESCs) generated from adult mouse RPE cells [22]. The functionality of these cells was studied in a mouse model of retinal degeneration. A neurosphere-induced reprogramming protocol can immortalize and transform mouse RPECs into iRPESCs with a dual potential to differentiate into cells that express either RPE or photoreceptor markers both in vitro and in vivo. iRPESCs, when subretinally transplanted into mice with retinal degeneration, can integrate into the RPE and neuroretina, thereby delaying retinal degeneration in the model animals. It is assumed that Hippo factor Yap1, a transcriptional activator, is upregulated in the nuclei of cells forming neurospheres. Factors Zeb1 and P300 (transcriptional coactivators) are downstream of the Hippo pathway and can bind to the promoters of stemness genes *Oct4*, *Klf4*, and *Sox2*, thereby transactivating them to reprogram RPE cells into iRPESCs [22]. The role of TFs possessing their properties of “pioneer” TFs, i.e., capable of inducing direct reprogramming of human RPE cells and obtaining differentiated RPE cells for regenerative cell therapy was shown in a few recent studies [397,398,399]. The potential for direct reprogramming of human RPE cells has been demonstrated in a number of papers [400,401,402,403,404,405,406,407].

It was suggested that the terminally differentiated mammalian RPE is rich in various epigenetic regulatory mechanisms that ensure a stable state of cell differentiation by rigidly fixing specific patterns of gene expression [408]. Hyperacetylation of core histones may lead to transcriptional reprogramming and activate the initial regenerative programs [409]. Further research has confirmed this assumption by providing data on more epigenetic alterations identified in aging cells, including loss of nucleosomes, altered histone modifications and levels, changes in DNA methylation, and non-coding RNA signatures [410].

EMT has vital significance in tissue healing and organ fibrosis and involves several known signaling cascades in RPE cells. Associations between EMT-associated TFs and epigenetic regulatory machinery provide possible routes to regulate the alterations in DNA methylation and histone modifications [411,412], which may have cell-specific outcomes [41]. Mapping the associated epigenetic and transcriptional changes in normal and EMT RPE may help elucidate the mechanisms of these processes [41]. The studies in this area raise the importance of documenting the changes at molecular and epigenetic levels in the RPE and adjacent eye tissues and, where possible, with unified approaches to enable more precise comparisons. Obviously, the observed reorganization of enhancer patterns is consistent with the RPE cells undergoing a global change in regulatory programs in response to the signaling from the cellular microenvironment.

The success of RPE cell reprogramming is largely determined by coordinated communication between signaling networks. The regulatory effects within the signaling pathways are not unidimensional but emerge as a result of the complex crosstalk and mutual influence of multiple factors. FGF2 is known among growth factors involved in initiating the earliest RPEC responses to retinal detachment in various vertebrates [19,238]. Proliferation of RPECs in the adult newt is carried out with the participation of FGF2, which is activated by a small delay relative to the time of induction of RPE cells to reprogramming when dissociated from the neural retina (NR). This demonstrates the main evolutionarily conserved mitogenic function of Fgf2 [167]. FGF2 has a crucial role in triggering RPE reprogramming in chicken embryos [238]. The results obtained on RPECs of chicken embryos confirmed the global rearrangements of DNA methylation patterns, including differentially methylated regions at promoters of genes associated with chromatin organization [235]. The important role of FGF2 is thought to support the rearrangements of DNA methylation patterns and the dynamic modifications during RPEC reprogramming to form a new NR in chicken embryos [235]. Recently, using this model, the dependence of RPE reprogramming on the changes in intracellular metabolism has been revealed. As that study has shown, the activation of retinol metabolism is concurrent with RPE fate restriction and vice versa and thus impacts cell fate decisions during RPE reprogramming [413]. This finding suggests that the metabolic pathways associated with RPE differentiation status may limit reprogramming competence through impact on epigenetic regulation.

The mammalian RPE cells have properties that are required for the initiation and progress of reprogramming along the neuronal and glial pathway, such as the proliferative activity and loss of melanin granules [414]. FGF2 is widely known as a stimulator of human RPE conversion and proliferation in vitro [33]. The studies on human RPE cell proliferation have shown that these cells implement the mechanism of S-phase entry as in the amphibian in vivo and avian RPE cells in vitro, where MAPK and ERK kinase pathways are used [19,33,238]. FGF2 binds to FGFR1 receptors on the cell surface, which activates MAPK/ERK and MMP9 expression, increases expression of Snail and Slug (SNAI2), mesenchymal genes, and decreases expression of epithelial genes in mammals [415]. Fgf2, critical for regeneration in lower vertebrates, has been implicated in compromising the RPE barrier function and precipitating degenerative diseases of the RPE in humans. Fgf2 can trigger cell proliferation and promote the manifestation of stem/progenitor cell properties, which are also signs of the development of EMT in mammals [416]. FGF2 can induce invasion processes during oncogenesis due to uncontrolled proliferation [417]. FGF2 is found in high concentrations in neovascular AMD. The increased secretion of FGF-containing vesicles stimulated by complement activation resulted in a decrease in RPE cell viability [418]. The modulation of RhoA/Rho kinase, Smad, and MAPK signaling involved in RPE pathology is considered in EMT treatment strategies [415,419]. The dysregulated RPE-EMT proteome shares commonality with malignancy-associated EMT. Altered RPE-EMT proteome protein signatures correlated with known AMD-associated risk factors were identified [420]. The mechanisms of FGFR1 activation by Fgf2 in RPE cells or using alternative receptors in EMT and in oncogenic transformation are not well studied. These processes are likely to be largely associated with a failure in epigenetic regulation, in addition to mutations of regulatory genes, that can lead to their aberrant expression. Thus, functions of Fgf2 in RPEC reprogramming in mammals are more complicated and demonstrate both significant and limited potential.

The susceptibility to cancer initiation and progression in adult mammals is speculated to be determined by a strong immune response and associated loss of an advanced regenerative capability during evolution [363]. Epigenetic mechanisms are essential for modulating gene expression and enabling the acquisition of cellular identity in embryogenesis and during cell reprogramming in regeneration and cancer. Mutations in components of the epigenetic machinery are another serious cause of neurodevelopmental and cancer disorders. These pathological consequences are associated with deficits in the epigenetic machinery and are considered a group of chromatinopathies closely associated with homeostasis imbalance [421,422].

In the concept of RPE reprogramming and retinal regeneration, the coordinated switch on the expression of genes allows the transition from a quiescent to a pro-regenerative form. The unique capacity of the newt to regenerate the retina after injury is governed by cell reprogramming of RPECs, proliferation, and progression. The critical way by which regeneration might suppress tumorigenesis is likely the greater competitiveness of regenerative cells than cancer cells to protect against unwanted spontaneous tumor transformation in highly regenerative species [364,365]. This assumption could possibly explain the success of RPE cell reprogramming in lower vertebrates, such as tailed amphibians.

However, the precise epigenetic mechanisms underlying this competition, the nuances of epigenetic remodeling during regeneration in different species, as well as the universality and the species-specific reprogramming factors, are yet to be explored. The accumulating comparative data also raise the task of further studying the evolution of signaling pathways in order to understand how the phenotypic switch inherent to the neural/retinal or mesenchymal phenotype is realized. How they are regulated at the epigenetic level remains to be elucidated in future studies.

## 9. Conclusions

Regulation of RPE conversion in regenerating and non-regenerating models has specificity, determining its strategy(s) and outcome: manifestation of regenerative capabilities or RPE-dependent retinal pathologies. Epigenetic switches, including DNA methylation and histone modifications, play pivotal roles in orchestrating RPEC responses through endogenous and exogenous modulators.

Even fragmentary data on epigenetic changes at the level of DNA and histone modifications, when comparing RPE-associated retinal diseases with RPE cell reprogramming into neurons in regeneration-competent animals, allow us to highlight the main differences (see Table 1, Figure 6). For amphibians and chicken embryos, the features revealed at the epigenetic level testify to a special regeneration response and behavior of RPECs. This strategy is aimed at triggering genes relevant to the formation of the eye in development, as well as multipotency genes. In RPEC reprogramming, the demethylation and acetylation of DNA, the histone modifications, and the modulation of embryonic histone expression play critical roles.

The changes in mammalian RPE cells under special in vitro conditions show some similarities to those in amphibian and avian RPE during in vivo conversion. We can find some similarities both in the triggers used in RPE conversion and in the expression of conserved “developmental” TFs, namely, TFs from the group of pluripotency inducers, and in the recruitment of a number of signaling networks. Consequently, there is reason to believe that certain molecular and epigenetic links in the regulatory machinery of RPE cell-type conversion were retained in the evolution of vertebrates.

Epigenetic regulation in RPE-associated retinal diseases occurs in different ways depending on the changes that take place in the tissue with different disease patterns. Homeostasis of RPE cells is a critical regulator of the epigenetic landscape in the pathogenesis of RPE-related disease. Macular and other retinal dystrophies have at their core problems with RPEC dysfunction and degeneration. As a result, one can observe deficiencies in the epigenetic machinery, which are unable to support the correct expression of genes that are supposed to support the normal homeostasis and function of RPECs and retinal cells. In the case of RPEC EMT occurring in PVR, as well as in diabetic and fibrotic retinopathy, there is a very different pattern of changes in epigenetic regulation, which primarily serves EMT, i.e., cell conversion in the mesenchymal direction. In the latter case, the cardinal change in gene expression constitutes, in fact, a reprogramming of expression. In general, in the case of RPE-dependent diseases (AMD, DR, and PVR) discussed above and those associated with cellular aging, the epigenetic changes demonstrate a deficient landscape leading to gene expression changes that are shaped by dysregulation of cell homeostasis. Regarding the epigenetic control of RPE plasticity in mammals, it is evident that RPE cells are deficient in the functioning of regulatory elements responsible for proliferation and the induction of reprogramming in neuronal/retinal directions. The findings demonstrate that active DNA demethylation coupled to histone modifications is a key process that may be applied to remove epigenetic barriers in order to regenerate type-specific retinal neurons in mammals.

However, despite the fact that the signaling molecules and TFs used to control the conversion of pigment epithelia of the eye exhibit evolutionary conservatism, they are organized in different regulatory networks and may be realized through different mechanisms. Comparative studies of the epigenetic mechanisms of RPEC conversion in species capable of endogenous regeneration of the RPE after injury present a unique opportunity to identify the epigenetic factors that may be potential targets for modulating RPE conversion in mammals in therapeutic treatment. A deeper understanding of epigenetic regulation is necessary to find approaches aimed at maintaining the stability of RPE cells and to restore their functions in retinal pathologies in humans. The chromatin state of RPEC-derived proliferating progenitor cells forming a population of retinal cell phenotypes, as well as epigenetic mechanisms of the RPE reprogramming at all its stages, requires further detailed studies. The accumulation of data concerning the differences between RPECs of the eye, in lower vertebrates during regeneration and in mammals during pathology, from a comparative perspective, may help in the search for new therapies for epigenetic disorders.

## Figures and Tables

**Figure 1 biomedicines-13-01552-f001:**
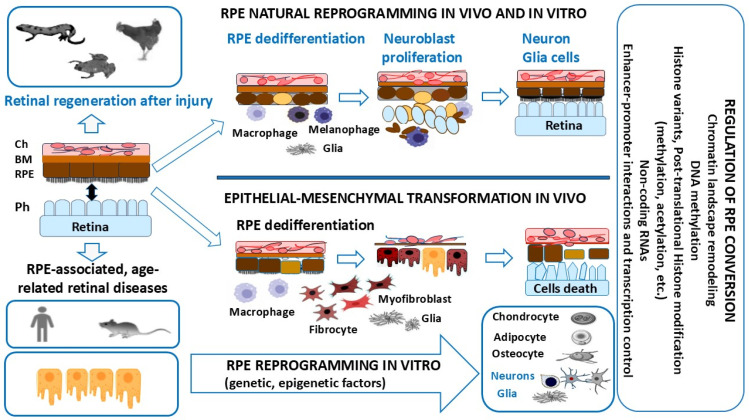
The strategies of RPE cell conversion after uncoupling RPECs’ interactions with retinal photoreceptors in vertebrates. RPECs are able to undergo genome reprogramming to produce all types of neurons and glia, and to regenerate the retina in lower vertebrates (amphibians and chicken embryos). RPECs undergo conversion into mesenchymal cell phenotypes during EMT in mammals and humans. Mammalian RPECs are capable of conversion into neurons under conditions of directed differentiation in vitro. Epigenetic and molecular regulation determines the species-specific phenotypic plasticity of RPE cells. More details are provided in the text.

**Figure 2 biomedicines-13-01552-f002:**
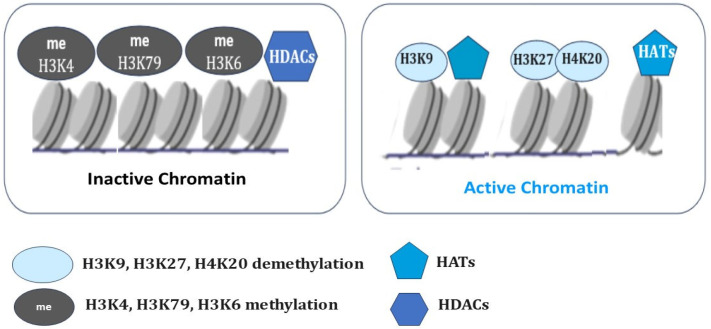
The selective histone modifications in the regulation of the chromatin state. Histone acetyltransferases (HATs) add an acetyl group, and histone deacetylases (HDACs) remove an acetyl group. Histone methylation (H3K9, H3K20, H3K27) and deacetylation (HDACs) result in chromatin condensation and transcriptional repression. Histone demethylation (H3K9, H3K27, H4K20) and acetylation (HATs) maintain the open chromatin state and allow gene transcription.

**Figure 3 biomedicines-13-01552-f003:**
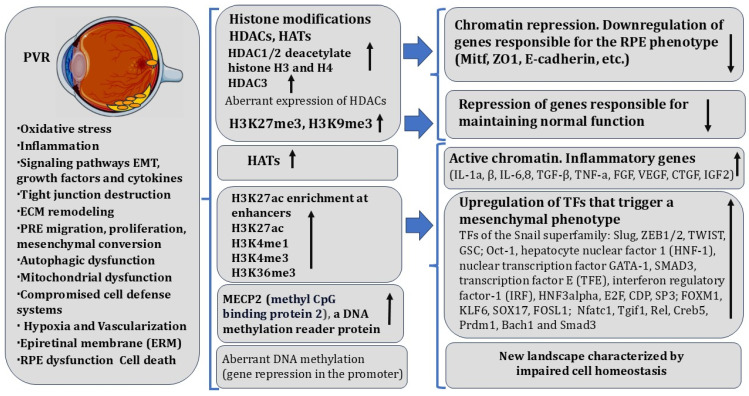
Gene-specific changes in the epigenetic landscape in RPE during proliferative vitreoretinopathy. The figure contains modified images from Servier Medical Art (https://smart.servier.com (accessed on 5 May 2025)) licensed by the Creative Commons Attribution CC BY 4.0 International License (https://creativecommons.org/licenses/by/4.0/ (accessed on 5 May 2025)). Arrows indicate activation/inhibition of the process. More details are provided in the text.

**Figure 4 biomedicines-13-01552-f004:**
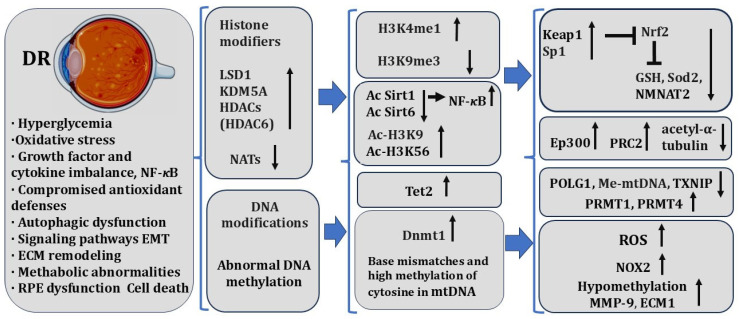
Epigenetic modifications in RPE during diabetic retinopathy. The figure contains modified images from Servier Medical Art (https://smart.servier.com (accessed on 5 May 2025)) licensed by the Creative Commons Attribution CC BY 4.0 International License (https://creativecommons.org/licenses/by/4.0/ (accessed on 5 May 2025)). Arrows indicate activation/inhibition of the process. More details are provided in the text.

**Figure 5 biomedicines-13-01552-f005:**
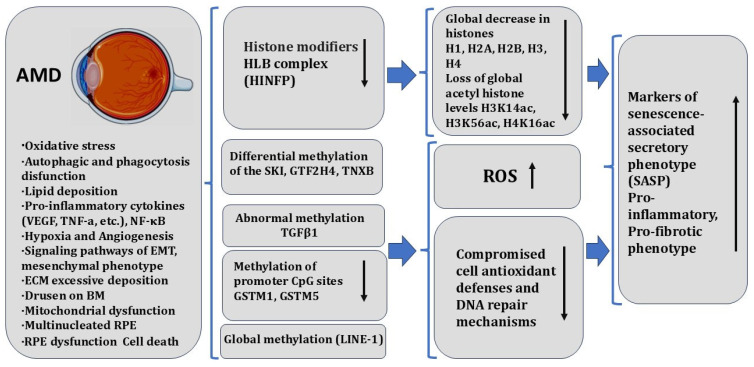
Epigenetic modification in RPE during age-related macular degeneration. The figure contains modified images from Servier Medical Art (https://smart.servier.com (accessed on 5 May 2025)) licensed by the Creative Commons Attribution CC BY 4.0 International License (https://creativecommons.org/licenses/by/4.0/ (accessed on 5 May 2025)). Arrows indicate activation/inhibition of the process. More details are in the text.

**Figure 6 biomedicines-13-01552-f006:**
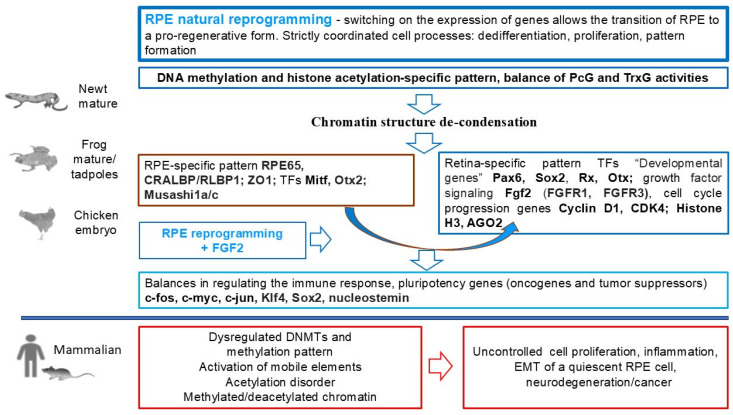
Summary scheme of the involvement of DNA methylation and histone acetylation in the initiation of the RPE cell conversion strategies.

**Figure 7 biomedicines-13-01552-f007:**
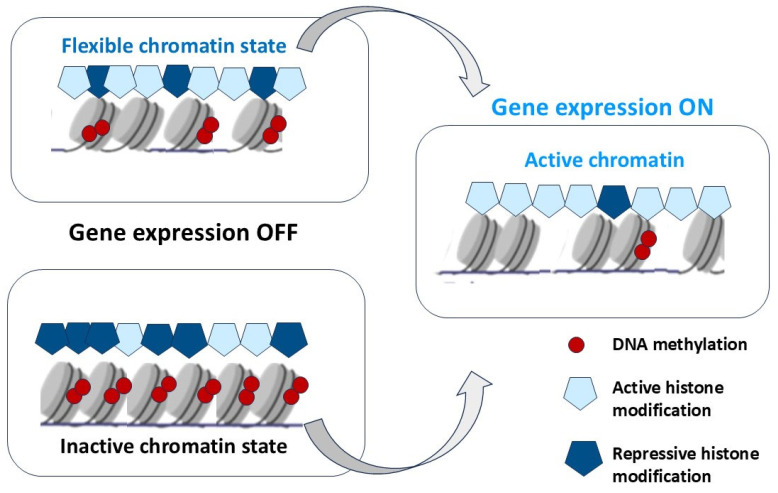
Hypothetical relationships between RPE cell reprogramming in caudate amphibians and chromatin state (DNA methylation and histone acetylation) on the regulation of gene transcriptional activity. In RPECs, the flexible chromatin has both active and repressive histone modifications and low DNA methylation, resulting in a silent state of target genes. Inactive chromatin is dominated by repressive histone marks and/or hypermethylated DNA, resulting in more closed chromatin. The switch in the transcriptional state from OFF to ON should be more feasible in flexible chromatin than in inactive chromatin.

**Table 1 biomedicines-13-01552-t001:** Shared and specific cellular, molecular, and epigenetic characteristics between retinal regeneration and pathology resulting from RPE cell conversion in lower and higher vertebrates.

Model Systems	Major Events in RPECs After Damage	Major Epigenetic Alterations in RPECs After Damage
**RPECS in tailed amphibians in vivo**	Cellular stress. Inflammation-related signaling pathways, permissive for retinal regeneration [153,161,167,168,237]. Reprogramming: Dedifferentiation, ECM remodeling [160,165]. Downregulation of genes, encoding RPE-specific markers; production of stem-like neuroblasts [18,152,159]. Remodeling of ribosome synthesis. Downregulation of protein translation processes. Nucleolar stress, activation of nucleolar genes [154,166,167]. Activation of pluripotency TFs [151,163]. and «Eye Field TFs» [19,155,157].	Chromatin remodeling [182]. Modulation of embryonic histone expression [217,219]. Activation of RNA-binding protein Musashi-1, recruitment of AGO2, a key effector of RNA silencing and modulator of chromatin remodeling [154]. Dynamics of nucleolar proteins [154,166,167]. Specific signal-response enhancers. Hypothesis: Hyperacetylation of core histones. TF promoters, responsible for pluripotency, are initially hypomethylated, indicating the “readiness” of the RPEC epigenome for dedifferentiation and reprogramming to produce retinal cells in regeneration-competent species.
**RPECs in tadpole *Xenopus laevis* in vivo/in vitro**	Cellular stress Signaling pathways, permissive for retinal regeneration [17,188]. Transdifferentiation. Dedifferentiation [199], Upregulation of MMPs, IL-1β, and TNF-α [195]. ECM remodeling [14]. Activation of pluripotency TFs [194] «Eye Field TFs» [192].	Chromatin remodeling. Modulation of embryonic histone expression [215]. DNA demethylation. Specific signal-response enhancers [192].
**RPECs in chicken embryos in vitro**	Cellular stress. Signaling pathways, permissive for retinal regeneration [238]. Reprogramming [235,236,237]. Dedifferentiation, ECM remodeling Activation of pioneer pluripotency TFs [238], and «Eye Field TFs» [235,238]. Metabolic changes affect RPE cell fate conversion	The dynamic balance between histone modifier enzymes [235]. Global changes in DNA methylation [234,235]. Demethylation of regulatory elements of photoreceptor genes [238]. Specific signal-response enhancers [234].
**ARPE19 in vitro**	Cellular stress (oxidative, nucleolus and nucleolar) [41,267]. Mitochondrial and lysosomal dysfunction [28]. Global transcriptome changes [252,253].	Chromatin remodeling. Changes in non-coding RNA signatures. Specific signal-response enhancers [41].
**Mouse RPECs (in vitro/in vitro)**	Cellular stress. Impaired *RPE* cell structure or function. Mitochondrial and lysosomal dysfunction. Signaling pathways promoting stemness genes [22].	Reduction in core and linker histone components H1, H2A, H2B, H3, and H4 [56]. Dysfunctional epigenetic landscape, deficient/aberrant character of epigenetic mechanisms, RPE senescence, and degeneration [40,41]. Location of most gene promoters of stem and progenitor cells in open chromatin [40]. Embryonic-like state of methylomes in adult RPE cells [22]. Demethylation of regulatory elements of photoreceptor genes [40].
**Human RPEs in vivo (AMD, DR, PVR)**	Cellular stress (oxidative, nucleolus, and nucleolar) [41]. Mitochondrial and lysosomal dysfunction Inflammation-related signaling pathways inhibiting retinal regeneration Impaired *RPE* cell structure or function, RPE senescence and degeneration [41]. ECM remodeling, formation of the ERM Expression of TFs specific for EMT, fibrosis/oncotransformation Diseases specific transcriptomic changes in RPE [2,56].	Dysfunctional epigenetic landscape [56], deficient/aberrant character of epigenetic mechanisms [40,41], abnormal methylation, and methylation in non-promoter regions Loss of nucleosomes. Loss of core and linker histones H1, H2A, H2B, H3, and H4 components [56]. Altered histone modifications and levels Changes in chromatin and DNA methylation and acetylation [55,56]. Diseases-specific DNA methylomes DNA demethylation in promoters of gene inducers of RPE diseases Specific signal-response enhancers [40,55].

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
