# Peer review of "Epigenetic Modifications in the Retinal Pigment Epithelium of the Eye During RPE-Related Regeneration or Retinal Diseases in Vertebrates"

_biomedicines, 2025, doi:10.3390/biomedicines13071552_

Round 1
Reviewer 1 Report
Comments and Suggestions for Authors
The topic of this manuscript is accurate and interesting, addressing an area of clear relevance to the field. The manuscript is well structured, with a logical sequence and clear main conclusions. However, a few minor changes are needed to further improve clarity and strengthen the overall impact of the work. I recommend addressing these minor points before publication:
- In chapter 6.1., first sentence "In human AMD, PVR, and DR are often caused by age-related aging of RPE cells." gives the impression that main pathophysiological process in diabetic retinopathy is linked to the age and maturing which is not correct. Remove this part of the text or define the difference in diseases stated.
-
The content of Table 1 is good, but the visual clarity is not. Put the statements in bullets or each in new line to make it visually more clear and understandable.
- The reference list lacks of recent publications on the current topic. There are some valid researches such as Zong et al. New perspectives on DNA methylation modifications in ocular diseases (doi: 10.18240/ijo.2025.02.19) or Chen et al. Alternative oxidase blunts pseudohypoxia and photoreceptor degeneration due to RPE mitochondrial dysfunction (doi: 10.1073/pnas.2402384121.) with current known data that could be appropriate for this topic.
Response to Reviewer 1
Dear Reviewer, thank you very much for taking the time to review our manuscript «Epigenetic modifications in the retinal pigment epithelium of the eye during RPE-related regeneration or retinal diseases in vertebrates» (Grigoryan E.N., Markitantova, Y.V.). The authors are grateful to Reviewer for editing the text and for the valuable recommendations.
Please find the detailed responses below and the corresponding revisions in track changes in the re-submitted files. Edited or added fragments are highlighted in grey color across the text.
- Response 1
We agree with this comment. The incorrectly formulated sentence has been removed from the text, and we also have added a fragment emphasizing the existence of differences in AMD, PVR and DR mechanisms (despite the existence of a number of common pathogenetic factors).
“In human, AMD, PVR and DR are multifactorial diseases that have distinct pathophysiological mechanisms, but ultimately impact the functionality of RPE cells. The dysfunction of RPE cells can exacerbate these diseases. RPE cells are known to undergo oxidative stress, and in pathologies they exhibit pro-inflammatory responses, mitochondrial and lysosomal dysfunction [274,275], loss of all mitochondrial electron transport complex (ETC) activities [276], nucleolus and nucleolar stress [277]. Homeostatic imbalance in RPECs is due to loss of proteostasis as a result of oxidative stress and senescence [278]. Understanding the molecular and epigenetic mechanisms that drive RPE aging is critical to our fundamental scientific understanding of cellular longevity and to developing novel effective therapies for age-related retinal diseases. In eye diseases, the inflammation, the oxidative stress and the lipid metabolism disorders are major pathogenic factors. The molecular-genetic factors mediating their action may be closely related to abnormal DNA methylation, and are being considered to be as the targets for epigenetic regulation for the therapy [279].”
Please, see: line 785-798
- Response 2
We agree with this comment. We have made a change to the table for visual clarity. The text was shortened and the statements were highlighted in bullets.
Please, see table 1, line 780
- Response 3
Thank you very much for the recommendations to pay attention to these interesting recent publications on the current topic. We have included in our manuscript the recommended papers, devoted to the study of mitochondrial dysfunction in RPE cells (Chen, M.; Wang, Y.; Dalal, R.; Du, J.; Vollrath, D. Alternative oxidase blunts pseudohypoxia and photoreceptor degeneration due to RPE mitochondrial dysfunction. PNAS. 2024, 121 (25) e2402384121. doi.org/10.1073/pnas.2402384121. [276]); and the deal with the relationship of epigenetic regulation (DNA methylation modifications) to the development of various eye diseases and new approaches to their therapy (Zong, F.F.; Jia, D.D.; Huang, G.K.; Pan, M.; Hu, H.; Song, S.Y.; Xiao, L.; Wang, R.W.; Liang, L. New perspectives on DNA methylation modifications in ocular diseases. Int J. Ophthalmol. 2025, 18 (2):340-350. doi: 10.18240/ijo.2025.02.19. [279]).
Please, see: line 788-790 [ref. 276]; line 794-798 [ref. 279].
Sincerely,
Authors
31 May 2025
Reviewer 2 Report
Comments and Suggestions for Authors
This comprehensive review article discusses the role of epigenetic modifications in RPE cells, comparing mechanisms of retinal regeneration in lower vertebrates with retinal pathologies in higher vertebrates, including humans. The paper covers DNA methylation, histone modifications, and the regulation of gene expression, linking them to RPE plasticity, dedifferentiation, and transdifferentiation. The authors provide a detailed comparison of regeneration-capable species versus disease-prone species, emphasizing the influence of epigenetic landscapes on cellular behavior. A few comments need to be addressed before publication.
- Section 4 focuses on iris pigment epithelial cell reprogramming, which is not clearly justified within the scope of the paper. Please clarify the relevance of this section to the central thesis of the manuscript.
- The manuscript would benefit from a thorough language revision. Several sections contain grammatical errors.
- FGF2 is described as a critical factor in promoting RPE reprogramming in regenerative species such as amphibians and chick embryos. However, its role in mammals (e.g., humans or mice) is not addressed. Given the comparative framework of the paper, it would be helpful to briefly discuss whether FGF2 has a known role — or limited potential — in mammalian RPE reprogramming.
- The manuscript presents a compelling comparison between the epigenetic regulation of RPE plasticity in regenerative species (e.g., amphibians and avians) versus non-regenerative species (e.g., mammals). However, the paper would benefit from a brief discussion of the evolutionary rationale behind this divergence. Why might mammals have evolved to epigenetically silence genes that promote RPE regeneration, in contrast to lower vertebrates that retain such plasticity? Additionally, it would be valuable to address the potential trade-offs or risks associated with reactivating these regenerative pathways in humans using genetic or molecular techniques — for example, possible links to oncogenic transformation or loss of cellular control.
Response to Reviewer 2
Dear Reviewer, thank you very much for taking the time to review our manuscript «Epigenetic modifications in the retinal pigment epithelium of the eye during RPE-related regeneration or retinal diseases in vertebrates» (Grigoryan E.N., Markitantova, Y.V.). The authors are grateful to Reviewer for editing the text and for the valuable recommendations.
Please find the detailed responses below and the corresponding revisions in track changes in the re-submitted files. Edited or added fragments are highlighted in grey color.
- Comments
Section 4 focuses on iris pigment epithelial cell reprogramming, which is not clearly justified within the scope of the paper. Please clarify the relevance of this section to the central thesis of the manuscript.
- Response 1
We agree with this comment. The review does indeed consider the topic of RPE cell reprogramming as the main one. However, we found it interesting and appropriate also to devote a chapter to the studies of epigenetic regulation in the iris pigment epithelium model (IRPCs). IRECs and RPECs have similar phenotypes and potential to reprogramming in lower vertebrates (Urodela) competent for retinal regeneration. Studies of both models (IRPCs and RPECs) are in favor of possible similarities between the epigenetic changes taking place in both cases. We have tried to emphasize and strengthen the rationale for including the iris cell reprogramming model at the beginning of Section 4.
Please, see: line 497-509
“The RPECs is an extension of the iris pigment epithelial cells (IPECs) by its embryonic origin and topologically, both have similar phenotypes. Therefore, the conversion of RPECs into retinal cells in Urodela is a model that is very close to the model of reprogramming iris pigment epithelial cells (IPECs) in lens cells during their regeneration in the same animals. Moreover, the IPECs, as the RPECs demonstrate the capacity to switch their phenotype in neural/retinal in vitro [13,20]. The process of pigmented epithelial cell reprogramming in RPECs and IPECs has similar regulatory molecular mechanisms. It is clear that direct extrapolation of the data obtained on the IPECs model is not entirely correct. Nevertheless, studies of both models are in favor of possible similarities between the epigenetic changes taking place in both cases. Thus, the analysis mechanisms of the different pigment epithelial cells reprogramming, makes it possible to understand the key factors that dictate a choice of particular cell-type conversion strategy. For this reason, we also present here the data obtained from the IPECs model. “
- Comments
The manuscript would benefit from a thorough language revision. Several sections contain grammatical errors.
- Response 2
We agree with this comment. The English translation of the text has been checked and the grammatical errors and typos have been corrected.
- Comments
FGF2 is described as a critical factor in promoting RPE reprogramming in regenerative species such as amphibians and chick embryos. However, its role in mammals (e.g., humans or mice) is not addressed. Given the comparative framework of the paper, it would be helpful to briefly discuss whether FGF2 has a known role — or limited potential — in mammalian RPE reprogramming.
- Response 3
We agree with this comment. Thank you for the recommendation. Fgf2 promotes RPE cell reprogramming and retinal regeneration in lower regeneration-competent species, but, indeed, has a limited role in this context in mammalian RPE. Moreover, in light of recent data, this growth factor in RPE is involved in signaling pathways that promote EMT. We have briefly discussed the role FGF2 in mammalian RPE, in comparison with RPE reprogramming in regeneration-competent species in Chapter Overall Discussion.
Please, see the included text: from line 1465 to line 1529
- Comments
The manuscript presents a compelling comparison between the epigenetic regulation of RPE plasticity in regenerative species (e.g., amphibians and avians) versus non-regenerative species (e.g., mammals). However, the paper would benefit from a brief discussion of the evolutionary rationale behind this divergence. Why might mammals have evolved to epigenetically silence genes that promote RPE regeneration, in contrast to lower vertebrates that retain such plasticity? Additionally, it would be valuable to address the potential trade-offs or risks associated with reactivating these regenerative pathways in humans using genetic or molecular techniques — for example, possible links to oncogenic transformation or loss of cellular control.
- Response 4
Thank you for the recommendation. These questions are really very interesting, important, and relevant, and are the subject of studies of many years of research on different models.
We have included the Chapter 7 “ENDOGENOUS PREREQUISITES DETERMINING RPE CELL CONVERSION, EPIGENETIC REGULATION” in manuscript.
In this Chapter we briefly reviewed the inherent features that ensure the successful RPE reprogramming in tailed amphibians, while in mammals, the inherent features block it and can lead to EMT and even the development of oncotransformation. The endogenic features in amphibians serve as prerequisites for regeneration and are determined by genetic regulatory networks, including signaling pathways and transcription factors, and by epigenetic machinery, the signals from the cellular microenvironment. The available information about them allows us to speculate that the reparative process in some caudate amphibians relies on molecular, epigenetic and cellular mechanisms capable of sensing abnormal (defective) signals and effectively reprogramming or overcome successfully these signals.
This emphasizes the importance and necessity of further study of questions concerning epigenetic regulation of the cell differentiation stability and plasticity, the relationship between the immune system regeneration, regeneration and carcinogenesis. It is likely that in tailed amphibians, evolutionarily formed features of endogenous regulation of RPE reprogramming along the neural pathway ensures the genomic stability and functionality, while minimizing the possibility of cell malignant transformation.
Please, see the included Chapter 7: from line 1100 to line 1238
Sincerely,
Authors
31 May 2025
Reviewer 3 Report
Comments and Suggestions for Authors
Reconsider after Major Revisions
The review by Grigoryan and Markitantova entitled “ Epigenetic Modifications in the Retinal Pigment Epithelium of the Eye During RPE-Related Regeneration or Retinal Diseases 3 in Vertebrates” is a very well-documented and well-written manuscript. This group has extensive experience writing reviews about similar topics.
In this review, the authors aim to compare epigenetic changes of the RPE during regeneration in different animal models as well as during disease in mammals including humans.
In general, the manuscript is very well organized and documented, the references are correct and accurate. However, the manuscript needs to be significantly improved before it is considered for publication.
This is a very extensive review and only ONE Figure and ONE Table are provided.
It will be very beneficial for the reader to summarize the different sections with Figures or Tables that can help to consolidate and understand the information provided.
Conclusion section:
Please include a Figure with more details of what is happening during RPE reprogramming b) RPE reprogramming during disease in mammals. What is common and what is different in terms of epigenetic changes or key factors.
Table 1: It’s confusing to the organization of this table; can you include horizontal lines? Are you mixing data from different models? Can you just include the epigenetic changes for each model. This is a significant amount of text for a Table.
Line 10: rephrase, in mammals including humans
Line 21: What means “deficient aberrant “pathological” epigenetic landscape? Please rephrase for clarity.
Linde 23: What means “epigenetic age maintaining” please rephrase for clarity.
Line 54: rephrase, in mammals including humans
Line 54: after retinal detachment and rupture, RPECs undergo
Line 66: Eliminate “and”
Line 77: Reference 50 is not related to the statements in the text. Please clarify or eliminate reference 50, what do you mean by methylation forms?
Line 300-308: What is the significance of reference 122 in terms of tissue regeneration? Please explain in more detail or eliminate.
Reference 199 incomplete
Please adjust format for reference 232.
Line 1275: Russian word please replace by english.
Response to Reviewer 3
Dear Reviewer, thank you very much for taking the time to review our manuscript «Epigenetic modifications in the retinal pigment epithelium of the eye during RPE-related regeneration or retinal diseases in vertebrates» (Grigoryan E.N., Markitantova, Y.V.). We express our deep gratitude to Reviewer for the editing and the valuable recommendations.
Please find the detailed responses below and the corresponding revisions in track changes in the re-submitted files. Edited or added fragments were highlighted in color across the text.
Reviewer 3
Submission Date
06 May 2025
Date of this review
20 May 2025 22:45:07
Comments and Suggestions for Authors
Reconsider after Major Revisions
The review by Grigoryan and Markitantova entitled “Epigenetic Modifications in the Retinal Pigment Epithelium of the Eye During RPE-Related Regeneration or Retinal Diseases 3 in Vertebrates” is a very well-documented and well-written manuscript. This group has extensive experience writing reviews about similar topics.
In this review, the authors aim to compare epigenetic changes of the RPE during regeneration in different animal models as well as during disease in mammals including humans.
In general, the manuscript is very well organized and documented, the references are correct and accurate. However, the manuscript needs to be significantly improved before it is considered for publication.
This is a very extensive review and only ONE Figure and ONE Table are provided.
It will be very beneficial for the reader to summarize the different sections with Figures or Tables that can help to consolidate and understand the information provided.
Conclusion section:
Please include a Figure with more details of what is happening during RPE reprogramming b) RPE reprogramming during disease in mammals. What is common and what is different in terms of epigenetic changes or key factors.
- Response
Thank you for the recommendations.
We have made and included Figure 2 in the Chapter 7 “Endogenous prerequisites determining RPE cell conversion, epigenetic regulation”, summarizing what a) happens during RPE reprogramming b) RPE reprogramming during disease in mammals. The figure shows the similarities and differences in terms of epigenetic changes and key factors.
and we have made Figure 3 which is included in the Chapter “Overall discussion” “Hypothetical relationships between RPE cell reprogramming in caudate amphibians and chromatin state (DNA methylation and histone acetylation) on the regulation of gene transcriptional activity.”
Comments
Table 1: It’s confusing to the organization of this table; can you include horizontal lines? Are you mixing data from different models? Can you just include the epigenetic changes for each model. This is a significant amount of text for a Table.
- Response
We agree with this comment. Therefore, we have given the name of the table. The idea of ​​the table emphasizes the shared and specific cellular, molecular and epigenetic characteristics between the retinal regeneration and pathology as a result of RPE cells conversion in lower and higher vertebrates. The contents of this table also have been organized for a visual clarity, and the text have been shortened, and highlight the statements in bullets.
Line 10: rephrase, in mammals including humans
- Phrase «in mammals and humans» were corrected to «in mammals including humans
Line 21: What means “deficient aberrant “pathological” epigenetic landscape? Please rephrase for clarity.
- We agree with this comment. Changed to “deficient epigenetic landscape”. This term is used to emphasize the epigenetic dysregulation, including DNA methylation, and histone modifications, the disorders in mechanisms. The word "pathological" is removed, since the term “deficit landscape” already implies disorders in regulation. The mutations as the components related to the epigenetic machinery also can lead to neurodevelopmental disorders and cancers. We also noted this in Section 7.
Please, see: line 21; line 1515
Linde 23: What means “epigenetic age maintaining” please rephrase for clarity.
- Phrase “epigenetic age maintaining” changed to “possibility of epigenetic maintaining the cellular identities” - line 24
Line 54: rephrase, in mammals including humans
- Сorrected to “in mammals including humans” - line 55
Line 54: after retinal detachment and rupture, RPECs undergo
- Added missing word “RPECs” - line 55
Line 66: Eliminate “and”
- Deleted “and”
Line 77: Reference 50 is not related to the statements in the text. Please clarify or eliminate reference 50, what do you mean by methylation forms?
- Response
Thank you for pointing this out. The text has been rewritten and a link to the relevant work has been provided.
“Temporal and cell type-specific expression of epigenetic modifiers and their selective interaction with a specific set of TFs ensure the sequential differentiation of retinal cells. Retina-specific epigenetic disorders in the DNA, and not only mutations, may contribute to the pathogenesis of a number of retinal diseases. Thus, irregularities in the DNA demethylation process of gene promoters and enhancers during proliferation and differentiation of retinal cell precursors into photoreceptors may significantly contribute to the development of retinitis pigmentosa (RP) [50].”
[50] Dvoriantchikova, G.; Lypka, K.R.; Ivanov, D. The Potential Role of Epigenetic Mechanisms in the Development of Retinitis Pigmentosa and Related Photoreceptor Dystrophies. Front. Genet. 2022, 13:827274. doi: 10.3389/fgene.2022.827274
Please see: from line 77 to line 83
Line 300-308: What is the significance of reference 122 in terms of tissue regeneration? Please explain in more detail or eliminate.
- Response
We agree with this comment. Thank you for pointing this out. The text has been rewritten and a link to the relevant work has been provided.
“Dnmt1 is also involved in the repression of retrotransposons (mobile elements) through DNA methylation in early development. This mechanism provides an additional stability for long-term repression and epigenetic propagation throughout development [121]. The ability of the retrotransposons to integrate into various genomic regions and trigger the isolation of promoters and enhancers disrupts their interactions, and also provides a mechanism associated with aberrant chromatin organization. During tissue regeneration, retrotransposon silencing is coupled with stem cell activity and essential for regeneration process: adult stem cells coordinately repress these mobile elements and activate lineage genes. Dysregulation of these processes, also related with risk of mutations, may initiate the developmental pathologies, including cancer [122].”
Please, see: line 309-319
Ref. [122] Lyu , Y.; Kim , S.J.; Humphrey , E.S.; Nayak, R.; Guan , Y.; Liang, Q.; Kim , K.H.; Tan, Y.; Dou, J.; Sun, H.; Song , X.; et al. Stem cell activity-coupled suppression of endogenous retrovirus governs adult tissue regeneration. Cell 2024, 187 (26):7414-7432.e26. doi: 10.1016/j.cell.2024.10.007.
Reference 199 incomplete
- Response
We agree. Thank you for pointing this out. The full reference is provided in the list Cited reference is provided [199]. Sato T. Uber die linsen-regeneration bei den Cobitiden Fischen Misgurnus Anguillicaudatus. Embryologia 1961, 6: 251–291.
Please adjust format for reference 232
- Response
Format have been adjusted for reference [232]
Line 1275: Russian word please replace by english.
- Replaced by english «allow»; Please, see: Line: 1540
Sincerely,
Authors
31 May 2025